# Goal-driven Bayesian Optimal Experimental Design for Robust Decision-Making Under Model Uncertainty

## Abstract

Bayesian optimal experimental design (BOED) aims to predict experiments that can optimally reduce the uncertainty in the model parameters. However, in many decision-critical applications, accurate parameter estimation does not necessarily translate to better decision-making, as not all parameters may significantly affect the efficacy of the decisions made in the presence of uncertainty. In this work, we propose GoBOED (Goal-driven Bayesian Optimal Experimental Design) to directly optimize the experimental design to reduce the model uncertainty that critically affects the quality of the downstream decision-making task of interest. We establish a computationally tractable connection between BOED and robust optimal control based on an uncertain model through convex optimization. This new integrated framework for robust control under uncertainty enables efficient gradient computation through a decision layer in GoBOED. Leveraging amortized variational inference, we create a differentiable pipeline that can identify optimal experiments targeting decision value. Unlike traditional information-maximizing designs, GoBOED can provide flexibility in experimental selection, as the experiment with the lowest data acquisition cost may be prioritized when multiple experiments lead to equivalent decision quality despite their difference in reducing the parameter uncertainty. The application of GoBOED to real-world problems, such as source localization, epidemic management and pharmacokinetic control, demonstrates the efficacy of our proposed goal-driven experimental design approach.

## 1 Introduction

When experiments are expensive, time-consuming, with potential safety risks, optimizing the experiment becomes crucial. For example, for systems identification in such scenarios, we need to carefully select the most informative experiments to accurately estimate the model parameters that govern the system. **Bayesian optimal experimental design (BOED)** provides a systematic framework specifically designed for this purpose, allowing researchers to identify maximally informative experimental designs (Chaloner & Verdinelli, 1995; Barber & Agakov, 2004; Lindley, 1956; Rainforth et al., 2024). BOED uses updated posterior distribution and compares this distribution with prior distribution. This approach has found applications across diverse fields including psychology (Bach, 2023), geophysics (Strutz & Curtis, 2024), and other domains where experimental resources are limited. A detailed discussion of related work on BOED is provided in Appendix G

In the operational research community, recent research has developed strategies for **robust decision-making under uncertainty**. For example, optimal control algorithms have been shown effective in epidemic management for developing policies that reduce both public health and economic impacts (Linde et al., 2009; Nowzari et al., 2016; Paré et al., 2020; Gardner et al., 2021). By focusing on the structure of the compartmental epidemiological models before evaluating the final output, computationally efficient strategies based on the Susceptible-Infected-Quarantined-Recovered (SIQR) model have been developed for optimal control of infectious disease while maintaining social functioning and minimizing disruptions under resource constraints (Ma et al., 2023; Ofir et al., 2022). Another real-world example is for quantifying how medicines are absorbed, distributed, metabolized, and eliminated in the body using Pharmacokinetic (PK) models (Mould & Upton, 2013; Zou et al.,

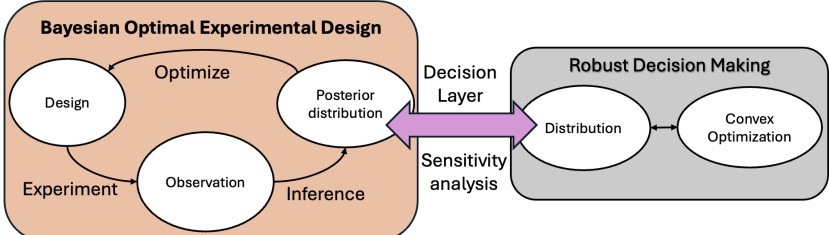

Figure 1: Interaction between Bayesian Optimal Experimental Design (BOED) and robust decision-making: In BOED (left), observations from an experiment update the posterior, and the (expected) posterior distribution then guides the design of the next experiment. The decision layer (center) maps the posterior to decision-relevant quantities (e.g., losses, risks, constraints) and supports sensitivity analysis of how decisions vary with uncertainty. In the robust decision-making module (right), a convex optimization uses this uncertainty representation to select actions that perform well under uncertainty. Our method couples these components by embedding the robust decision problem within the BOED loop: sensitivity analysis identifies regions of the posterior most relevant for decisions, and the resulting robust decisions are used to prioritize the next experiment.

2020). Within PK modeling, dose optimization is critical to maintain efficacy while minimizing adverse toxicity (Silva et al., 2025). However, the parameters in these models are inherently uncertain, making accurate estimation critical for effective decision-making. Vitková et al. (2023) addressed parameter uncertainty by analyzing open-loop control cycles. While robust control methods can accommodate parameter uncertainty and more general model uncertainty (Nemirovski, 2012), they often lead to overly conservative policies.

This paper combines two methods into one. We reduce the uncertainty of the model parameters by experiments and then use the updated distribution to compute the robust optimal control. Considering experimental design for the final operational goal of more effective decision-making under uncertainty, we propose a new **Goal-driven BOED (GoBOED)** framework to strategically design experiments to reduce model uncertainty that most significantly affects decision outcomes. We utilize variational inference methods to approximate accurate posterior distributions (Foster et al., 2019), while employing convex optimization methods for optimal control (Talaei et al., 2024; Ambikapathi et al., 2015). Figure 1 visualizes our proposed GoBOED framework.

Our GoBOED has maintained the interpretability of the solution. At the optimal point, we can perform Lagrangian sensitivity analysis and use the derivative information to guide experimental design. This allows us to efficiently identify informative experimental designs that have the greatest impact on decision-making processes, creating a more direct bridge between experimental observation and control implementation in convex settings. Moreover, GoBOED significantly reduces computational cost by amortizing inference: we train a single attention-based variational network once to learn the variational posterior and its associated gradient information, and then reuse this network during design optimization without repeatedly solving the full forward model.

We summarize our main contributions in this paper:

- We propose GoBOED to integrate Bayesian optimal experimental design with robust optimal control under uncertainty governed by convex optimization, enabling effective and efficient robust decision-making under uncertainty.
- We develop computational strategies that efficiently train posterior distributions while simultaneously making decisions under uncertainty.
- The proposed method is applicable to a broader class of real-world control problems formulated as convex optimization problems with uncertainty-aware complex system modeling, with demonstrated performances in epidemic management and pharmacokinetic control.

## 2 BACKGROUND

For clarity, we have provided a list summarizing the adopted mathematical notations and symbols in this paper with explanations in Table 1 of Appendix A.

## 2.1 Bayesian optimal experimental design (BOED)

BOED is an information-theoretic approach to the problem of identifying which experiments are most informative. It consists of a prior assumption about unknown model parameters $\boldsymbol{\theta} \in \Theta$, design variables $\xi \in \Xi$, a forward model $f : \Theta \times \Xi \to Y$, and observations $y \in Y$ given by $y = f(\boldsymbol{\theta}, \xi) + \epsilon$, where $\epsilon$ denotes noise, e.g., $\epsilon \sim \mathcal{N}(0, I)$.

Under this setting, we evaluate experimental designs by computing the expected information gain (EIG). This involves calculating the KL divergence between the posterior and prior distributions, and taking the expectation of this value with respect to the marginal likelihood. This formulation can be reformulated to estimate EIG by sample average approximation (SAA) using Bayes' rule:

$$\text{EIG}(\xi) := \mathbb{E}_{p(\boldsymbol{\theta})p(y|\boldsymbol{\theta},\xi)} \left[ \log \frac{p(y \mid \boldsymbol{\theta}, \xi)}{p(y \mid \xi)} \right], \tag{1}$$

with the most informative experiment by solving $\xi^* = \arg\max_\xi \text{EIG}(\xi)$.

Since computing the posterior distribution and KL divergence between two distributions can be computationally expensive, EIG has been often efficiently estimated using SAA with approximated posterior distributions. Recent work has introduced flexible neural approaches for this purpose, including normalizing flows (Dong et al., 2025; Orozco et al., 2024), diffusion-based generative models (Iollo et al., 2025), and mutual-information neural estimation techniques (Kleinegesse & Gutmann, 2020). These methods provide highly expressive approximations, albeit sometimes at considerable computational and implementation cost. As our paper focuses on real-world scientific problems and decision-aware optimization, we employ variational inference (Foster et al., 2019; 2020). A key benefit of this approach is that EIG with variational inference asymptotically converges to the true value as the number of samples increases, ensuring the soundness of decision outputs alongside computational efficiency.

When using variational inference, we can approximate the EIG by:

$$\text{EIG}(\xi) \leq \mathbb{E}_{p(\boldsymbol{\theta})p(y|\boldsymbol{\theta},\xi)} \left[ \log p(y \mid \boldsymbol{\theta}, \xi) - \mathbb{E}_{q_\phi(\boldsymbol{\theta}|y,\xi)} \left[ \log \frac{p(y \mid \boldsymbol{\theta}, \xi)p(\boldsymbol{\theta})}{q_\phi(\boldsymbol{\theta} \mid y, \xi)} \right] \right], \tag{2}$$

where $q_\phi(\boldsymbol{\theta} \mid y, \xi)$ is an approximated posterior distribution, and $\phi$ represents neural network parameters for the variational inference network, which generates the posterior parameters based on observation and design.

## 2.2 From BOED to Robust Decision-Making Under Uncertainty

In many real-world applications, optimal decisions depend on uncertain model parameters $\boldsymbol{\theta}$. Standard robust decision-making accounts for this uncertainty when choosing an action, but here we emphasize a complementary lever: *BOED* to actively reduce the uncertainty that matters for downstream decisions.

We first formalize the robust decision problem given a posterior. After running a design $\xi$ and observing $y$, we obtain a posterior distribution over parameters $\boldsymbol{\theta}$, $p(\boldsymbol{\theta} \mid y, \xi)$. Let $J(\boldsymbol{g}; \boldsymbol{\theta})$ denote the application cost corresponding to decision-making $\boldsymbol{g}$ given inferred model parameters $\boldsymbol{\theta}$ (equivalently, $J(\boldsymbol{g} \mid y, \xi) = \mathbb{E}_{\boldsymbol{\theta} \sim p(\boldsymbol{\theta}|y,\xi)}[J(\boldsymbol{g}; \boldsymbol{\theta})]$). We seek a decision that is robust with respect to the updated posterior:

$$\boldsymbol{g}^* \in \arg\min_{\boldsymbol{g} \in \mathcal{G}} \rho_{\boldsymbol{\theta} \sim p(\boldsymbol{\theta}|y,\xi)}[J(\boldsymbol{g}; \boldsymbol{\theta})], \tag{3}$$

where $\rho$ is a risk functional (e.g., expectation or chance constraints).

We further choose the design that minimizes the expectation of the robust decision problem before observing $y$. The posterior—hence the robust action in eq. (3)—improves in the directions that matter for $J(\boldsymbol{g}; \boldsymbol{\theta})$. The combining objective is

$$\xi^* \in \arg\min_{\xi \in \Xi} \mathbb{E}_{p(y|\xi)} \left[ \min_{\boldsymbol{g} \in \mathcal{G}} \rho_{\boldsymbol{\theta} \sim p(\boldsymbol{\theta}|y,\xi)}[J(\boldsymbol{g}; \boldsymbol{\theta})] \right], \tag{4}$$

which is decision-focused. We compute the expectation over marginal likelihood to make it general to possible observation. Equation (4) prioritizes experiments that most reduce eventual robust loss, not just overall parameter uncertainty. We illustrate this framework on three applications: source localization, epidemic management, and pharmacokinetic control. Detailed model specifications and solver settings for each example are provided in appendix C and appendix E.

## 3 METHODS

We develop an integrated framework, GoBOED, for goal-driven experimental design by bringing together BOED (introduced in eq. (2)) and robust optimal control under model uncertainty through convex optimization (described in eq. (3)). Our primary objective is to identify an optimal experimental design $\xi^*$ that minimizes the expected controlled economic cost (i.e., the control objective), where the expectation is taken over posterior distributions of the model parameters $\boldsymbol{\theta}$. This approach allows us to update our beliefs with new observations and improve decision-making regarding goal-driven optimal experiment and robust optimal control based on an uncertain model, following the methodology of Chaloner & Verdinelli (1995).

### 3.1 ROBUST DECISION-MAKING

We begin by formulating a robust control problem that explicitly accounts for uncertainty in the model parameters through an approximate posterior distribution. This formulation is central to GoBOED: it defines the decision-making objective that experiments are meant to improve. In particular, it allows us to optimize experimental designs not merely for accurate parameter estimation, but for the quality of the resulting control decisions. For a given experimental design $\xi$ and observed data $y$, we formulate the optimization problem as:

$$\min_{\boldsymbol{g}} \ \mathbb{E}_{p(\boldsymbol{\theta}|y,\xi)}[J(\boldsymbol{g}) \quad \text{s.t. } \text{Constraints}(\boldsymbol{g}, \boldsymbol{\theta})]. \tag{5}$$

Here, $J(\boldsymbol{g})$ is the control cost, assumed convex in $\boldsymbol{g}$, and it does not depend on $\boldsymbol{\theta}$ directly; uncertainty enters through the constraints in eq. (5). Under this convexity assumption we can carry out sensitivity analysis of the optimizer with respect to $\boldsymbol{g}$. The uncertain model parameters are collected in $\boldsymbol{\theta} \sim p(\boldsymbol{\theta} \mid y, \xi)$, and the key challenge is evaluating the posterior quantities such as probabilities of constraint satisfaction (cf. the chance-constraint formulation below). We address this using importance sampling with a variational proposal as detailed below.

To approximate the posterior distribution, we employ stochastic variational inference. Specifically, we approximate the true posterior $p(\boldsymbol{\theta} \mid y, \xi)$ using a variational distribution $q_\phi(\boldsymbol{\theta} \mid y, \xi)$, where $\phi$ denotes the variational parameters. The variational distribution is optimized by maximizing the evidence lower bound (ELBO), which can be expressed as:

$$\mathcal{L}_{\text{VI}}(\phi; y, \xi) = \mathbb{E}_{q_\phi(\boldsymbol{\theta}|y,\xi)} \left[ \log p(y \mid \boldsymbol{\theta}, \xi) \right] - D_{KL} \left( q_\phi(\boldsymbol{\theta} \mid y, \xi) \| p(\boldsymbol{\theta}) \right). \tag{6}$$

Once we have obtained the variational posterior $q_\phi(\boldsymbol{\theta} \mid y, \xi)$, we use it to estimate the expectation of the constraints. Since $q_\phi$ may not exactly match the true posterior, we apply importance sampling to correct for the discrepancy. For any function $f(\boldsymbol{\theta})$, the expectation under the true posterior can be estimated as

$$\mathbb{E}_{p(\boldsymbol{\theta}|y,\xi)}[f(\boldsymbol{\theta})] \approx \frac{\sum_{i=1}^N f(\boldsymbol{\theta}_i) w(\boldsymbol{\theta}_i)}{\sum_{i=1}^N w(\boldsymbol{\theta}_i)}, \tag{7}$$

where $(\boldsymbol{\theta}_i)$ are samples from $q_\phi(\boldsymbol{\theta} \mid y, \xi)$, and $w(\boldsymbol{\theta}_i) = \frac{p(y|\boldsymbol{\theta}_i,\xi)p(\boldsymbol{\theta}_i)}{q_\phi(\boldsymbol{\theta}_i|y,\xi)}$ are importance weights.

In particular, for the constraint, we approximate the posterior mean $\bar{\boldsymbol{\theta}}$ using the above formula with $f(\boldsymbol{\theta}) = \boldsymbol{\theta}$, and use this mean in the constraint. This allows us to evaluate the constraint and solve the optimization problem for each $\xi$ and $y$. The resulting formulation, denoted $\mathcal{L}_{\text{ROC}}$, is

$$\min_{\boldsymbol{g}} \ J(\boldsymbol{g}) \quad \text{s.t. } \text{Constraints}(\boldsymbol{g}, \bar{\boldsymbol{\theta}}) \tag{8}$$

where $\mathcal{L}_{\text{ROC}}$ can be viewed as a deterministic approximation of the original robust decision-making problem, obtained by enforcing the constraint only at the posterior mean parameter $\bar{\boldsymbol{\theta}}$. While this mean-based formulation is computationally convenient, it does not explicitly control the probability of constraint violation under the posterior distribution of $\boldsymbol{\theta}$.

### 3.1.1 ALTERNATIVE FORMULATION USING CHANCE CONSTRAINTS

To better incorporate the uncertainty in parameters $\boldsymbol{\theta}$ for robust optimization, we apply an alternative formulation using chance constraints, a powerful tool in optimization under uncertainty ensuring

that critical conditions hold with a specified probability (Charnes & Cooper, 1959; Miller & Wagner, 1965). Given that the objective function is independent of $\boldsymbol{\theta}$ in our formulations, we can concentrate on the constraints with posterior samples $\boldsymbol{\theta}_i \sim p(\boldsymbol{\theta} \mid y, \xi), i = 1, ..., N$. Directly imposing constraints for each sample would be overly restrictive. Instead, we introduce a chance constraint that requires the stability condition to be satisfied with high posterior probability under $p(\boldsymbol{\theta}|y, \xi)$.

The alternative formulation, denoted $\mathcal{L}_{\text{ROC-CC}}$, is defined as:

$$\min_{\boldsymbol{g}} \ J(\boldsymbol{g}) \quad \text{s.t.} \ \ \mathbb{P}(\text{Constraints}(\boldsymbol{g}, \boldsymbol{\theta}) \mid y, \xi) \geq \eta \tag{9}$$

where $\boldsymbol{\theta}$ follows the posterior distribution $p(\boldsymbol{\theta} \mid y, \xi)$ with the desired probability level $\eta \in (0, 1)$. We refer to $\mathcal{L}_{\text{ROC-CC}}$ as the *chance-constrained* robust decision-making objective.

To evaluate the chance term in eq. (9) efficiently, we use importance sampling with a variational proposal. Specifically, we set $f(\boldsymbol{\theta}) = \mathbf{1}\{\text{Constraints}(\boldsymbol{g}, \boldsymbol{\theta}) \mid y, \xi\}$ in eq. (7) to estimate $\mathbb{P}(\text{Constraints}(\boldsymbol{g}, \boldsymbol{\theta}) \mid y, \xi)$ stably.

1. Draw $N$ samples $\{\boldsymbol{\theta}_i\}_{i=1}^N$ from $q_\phi(\boldsymbol{\theta} \mid y, \xi)$.

2. Compute importance weights, and normalize it $w_i = \frac{p(\boldsymbol{\theta}_i) \, p(y|\boldsymbol{\theta}_i, \xi)}{q_\phi(\boldsymbol{\theta}_i|y, \xi)}, \tilde{w}_i = \frac{w_i}{\sum_{j=1}^N w_j}$.

3. Estimate the posterior probability (given $y, \xi$):
   $\mathbb{P}(\text{Constraints}(\boldsymbol{g}, \boldsymbol{\theta}) \mid y, \xi) = \sum_{i=1}^N \tilde{w}_i \, \mathbf{1}\{\text{Constraints}(\boldsymbol{g}, \boldsymbol{\theta}) \mid y, \xi\}$.

Using these importance-weighted samples, we impose the constraints and solve the resulting optimization via convex optimization.

### 3.1.2 Robust control using Conditional Value-at-Risk (CVaR)

As an alternative to the scenario-based chance constraint in Section 3.1.1, we control the *tail* of constraint violations using Conditional Value-at-Risk (CVaR).

Let the feasibility event be $\mathbb{P}(\text{Constraints}(\boldsymbol{g}, \boldsymbol{\theta}) \mid y, \xi) \geq \eta$. For samples $\{\boldsymbol{\theta}_i\}_{i=1}^N$ from the posterior distribution, define the per-sample violation $v_i(\boldsymbol{g}; \boldsymbol{\theta}_i)$. We enforce

$$\text{CVaR}_\eta\big(v(\boldsymbol{g}; \boldsymbol{\theta})\big) \leq 0$$

using the Rockafellar–Uryasev sample-average form with optional normalized weights $\bar{w}_i$:

$$s_i \ \geq \ v_i(\boldsymbol{g}; \boldsymbol{\theta}_i) - \tau, \quad i = 1, \ldots, N, \qquad \tau + \frac{1}{1-\eta} \sum_{i=1}^N \bar{w}_i \, s_i \ \leq \ 0,$$

with decision variables $\tau \in \mathbb{R}$ and $s_i \geq 0$. At optimality, $s_i = (v_i(\boldsymbol{g}; \boldsymbol{\theta}_i) - \tau)_+$, so only the upper tail beyond the $\eta$-quantile contributes.

The resulting robust optimal decision-making problem, denoted $\mathcal{L}_{\text{ROC-CVaR}}$, is

$$\min_{\boldsymbol{g}} J(\boldsymbol{g}) \quad \text{s.t.} \ \ \text{CVaR}_\eta\big(v(\boldsymbol{g}; \boldsymbol{\theta})\big) \leq 0 \tag{10}$$

and we refer to $\mathcal{L}_{\text{ROC-CVaR}}$ as the *CVaR-based* robust decision-making objective.

With these formulations in place, we can compute the robust optimal control under the approximate posterior distribution. The resulting optimal value of the control objective $J(\boldsymbol{g}^*)$ is precisely the quantity that GoBOED seeks to reduce: our experimental designs are chosen to minimize this robust control cost in expectation. In summary, $\mathcal{L}_{\text{ROC}}$, $\mathcal{L}_{\text{ROC-CC}}$, and $\mathcal{L}_{\text{ROC-CVaR}}$ denote the mean-based, chance-constrained, and CVaR-based robust decision-making objectives, respectively.

### 3.2 Differentiable Decision layer

We now connect the robust control formulation to the experimental design problem by viewing it as a *differentiable decision layer* inside GoBOED. Intuitively, this layer maps model parameters $\boldsymbol{\theta}$ to a robust control decision and its associated cost; by differentiating through this map, we can propagate

gradients back to the design variables $\xi$. A detailed derivation is given in Appendix D for the SIQR example, and the same construction applies to other models.

Concretely, for a given realization of $\boldsymbol{\theta}$, sampled from the posterior, the decision layer solves the convex robust control problem and returns the optimal solution $\boldsymbol{g}^*$ and objective value $J(\boldsymbol{g}^*)$, together with the KKT multipliers $\boldsymbol{\lambda}^*$ of the active constraints. We denote the optimal value by $J^*(\boldsymbol{\theta}; y, \xi) := J(\boldsymbol{g}^*(\boldsymbol{\theta}; y, \xi))$, which coincides with the robust decision-making objective for any of the formulations (we write $\mathcal{L}^*_{\text{ROC}}$, $\mathcal{L}^*_{\text{ROC-CC}}$, or $\mathcal{L}^*_{\text{ROC-CVaR}}$ when we need to distinguish them). Since the stage cost $J$ does not depend explicitly on $\boldsymbol{\theta}$ in our formulations, KKT-based sensitivity analysis yields

$$\nabla_{\boldsymbol{\theta}} J^* = \sum_{i \in \mathcal{A}} \lambda_i^* \, \nabla_{\boldsymbol{\theta}} l_i(\boldsymbol{g}^*, \boldsymbol{\theta}; y, \xi),$$

where $l_i(\boldsymbol{g}, \boldsymbol{\theta}; y, \xi) \leq 0$ are the constraints and $\mathcal{A}$ is the active set.

With this decision layer, the total derivative with respect to the design $\xi$ follows by the chain rule:

$$\frac{dJ^*}{d\xi} = \left(\nabla_{\boldsymbol{\theta}} J^*\right)^\top \frac{\partial \boldsymbol{\theta}}{\partial \xi},$$

where the Jacobian $\partial \boldsymbol{\theta} / \partial \xi$ is obtained via the reparameterization $\boldsymbol{\theta} = h(\epsilon; y, \xi, \phi)$ with $\epsilon \sim p(\epsilon)$ (see Section 3.4 for details). Any explicit dependence of the constraints on $\xi$ is handled via the terms $\nabla_\xi l_i(\boldsymbol{g}, \boldsymbol{\theta}; y, \xi)$.

The decision layer solves the inner robust control problem once (forward) and differentiates it via the KKT system (backward), making the robust decision step directly compatible with end-to-end training of GoBOED. For implementation, we use `cvxpylayers` (Agrawal et al., 2019) to map the convex optimization problem directly to its differentiable solution.

### 3.3 FORMULATION OF THE GoBOED END-TO-END OPTIMIZATION PROBLEM

By combining convex optimization with stochastic variational inference as described in the previous sections, we obtain a robust decision-making framework under parameter uncertainty with tractable gradients. Our next objective is to determine the optimal experimental design $\xi^*$ and variational parameters $\phi^*$ that jointly (i) minimize the expected cost of decision-making and (ii) ensure an accurate approximation to the posterior distribution of the parameters $\boldsymbol{\theta}$.

Because the posterior $p(\boldsymbol{\theta} \mid y, \xi)$ depends on a specific observation $y$, we follow Krishnan & Tickoo (2020); Lacoste–Julien et al. (2011) and compute expectations with respect to the marginal likelihood $p(y \mid \xi)$. We formulate the joint optimization problem as

$$(\xi^*, \phi^*) = \arg \min_{\xi \in \Xi, \, \phi} \mathbb{E}_{p(y|\xi)} \left[ \mathcal{L}^{(r)}_{\text{ROC}}(\boldsymbol{\theta}; y, \xi, \phi) - \mathcal{L}_{\text{VI}}(\phi; y, \xi) \right], \tag{11}$$

where $\mathcal{L}_{\text{VI}}$ is the variational objective from eq. (6), and $\mathcal{L}^{(r)}_{\text{ROC}}$ denotes the robust decision-making loss for a chosen robust control formulation $r$. In particular, $\mathcal{L}^{(r)}_{\text{ROC}} \in \left\{ \mathcal{L}_{\text{ROC}}, \mathcal{L}_{\text{ROC-CC}}, \mathcal{L}_{\text{ROC-CVaR}} \right\}$, corresponding to the mean-based, chance-constrained, and CVaR-based objectives, respectively. The minus sign in front of $\mathcal{L}_{\text{VI}}$ reflects that we simultaneously *minimize* the robust decision-making loss and *maximize* the ELBO, so that the posterior approximation is as accurate as possible while optimizing downstream decision quality. For notational simplicity, we now fix a particular robust control formulation $r$ and drop the superscript, writing $\mathcal{L}_{\text{ROC}}$ instead of $\mathcal{L}^{(r)}_{\text{ROC}}$ throughout the remainder of the paper.

In our implementation we adopt a simpler and more practical two-stage procedure: we first train the amortized variational network offline by maximizing $\mathcal{L}_{\text{VI}}(\phi; y, \xi)$ (Section 3.1), obtaining $\phi^*$. We then freeze $\phi$ and optimize only the design variable $\xi$ by minimizing $\mathbb{E}_{p(y|\xi)}\left[\mathcal{L}_{\text{ROC}}(\boldsymbol{\theta}; y, \xi, \phi^*)\right]$, as summarized in Algorithm 1.

#### 3.3.1 GRADIENT ESTIMATION

To optimize $\phi$, we use the reparameterization trick (Burda et al., 2015; Foster et al., 2020) to stabilize training and enable differentiation of the variational objective with respect to $\phi$. We maximize eq. (6)

by writing $\boldsymbol{\theta}_i = h(\epsilon_i; y, \xi, \phi)$, where $h$ is the variational encoder and $\epsilon_i \sim p(\epsilon)$, a standard normal distribution. The gradient is then approximated via Monte Carlo estimation as in Foster et al. (2019).

We set our robust decision–making loss function $L(\xi) = \mathbb{E}_{p(y|\xi)}[\mathcal{L}_{ROC}(\boldsymbol{\theta}; y, \xi, \phi)]$. By chain rule, the gradient with respect to $\xi$ can be written in the following forms,

$$
\frac{\partial L}{\partial \xi} = \mathbb{E}_{p(y|\xi)} \left[ \frac{\partial \mathcal{L}_{\text{ROC}}(\boldsymbol{\theta}; y, \xi, \phi)}{\partial \xi} + \mathcal{L}_{\text{ROC}}(\boldsymbol{\theta}; y, \xi, \phi) \frac{\partial \log p(y \mid \xi)}{\partial \xi} \right] \tag{12}
$$

The first term, $\frac{\partial \mathcal{L}_{\text{ROC}}(\boldsymbol{\theta}; y, \xi, \phi)}{\partial \xi}$, can be computed using implicit differentiation through decision layer, as described in Section 3.2. And the second term, $\mathcal{L}_{\text{ROC}}(\boldsymbol{\theta}; y, \xi, \phi) \frac{\partial \log p(y|\xi)}{\partial \xi}$, is straightforward to compute once an observation model (e.g., Poisson or Gaussian noise) is specified as described in Appendix D.4.

Together, these gradient estimators enable end-to-end optimization of both the experimental design $\xi$ and the variational parameters $\phi$ under the robust decision-making objective. A high-level summary of the full GoBOED pipeline, combining the amortized inference phase with the decision-focused design loop, is given in Algorithm 1 in Appendix B.1.

### 3.4 Optimization of experimental design

When the forward model is expensive to solve, recomputing a variational posterior at every candidate design is prohibitive. Motivated by Huang et al. (2024), we train a single amortized variational network that maps a design–observation pair $(\xi, y)$ to the parameters of a posterior $q_\phi(\boldsymbol{\theta} \mid \xi, y)$ in one shot and then use this network for gradient-based design optimization. This enables highly efficient experimental design.

**Amortized VI (one-shot training).** The amortizer takes $(\xi, y)$ as input, lifts them into a shared latent space of width $d$, and fuses the signals with a single-head attention block (queries from $y$, keys from $\xi$, values from both). This yields a variational posterior $q_\phi(\boldsymbol{\theta} \mid \xi, y)$ (see Appendix B.2 for the architecture). We train $\phi$ once using simulated trajectories $\{\xi_t, y_t\}_{t \in \mathcal{T}}$ and maximize the summed ELBO,

$$
\hat{\mathcal{L}}_{\text{VI}}(\phi) = \frac{1}{|\mathcal{T}|} \sum_{t \in \mathcal{T}} \mathbb{E}_{\boldsymbol{\theta} \sim q_\phi(\cdot|\xi_t, y_t)} \Big[ \log p(y_t \mid \boldsymbol{\theta}, \xi_t) + \log p(\boldsymbol{\theta}) - \log q_\phi(\boldsymbol{\theta} \mid \xi_t, y_t) \Big],
$$

which corresponds to eq. (6) evaluated across all $t \in \mathcal{T}$. The attention layer aggregates information across multiple designs without discarding earlier contributions, avoiding reliance on a fixed handcrafted summary.

**Design gradients via reparameterization.** After training, the amortizer produces variational parameters $\psi_\phi(\xi, y)$ for a family $q_\phi(\boldsymbol{\theta} \mid \xi, y)$. Assuming the family is reparameterizable, samples can be written as a deterministic transformation of base noise: $\boldsymbol{\theta}_s(\xi, y) = h_\phi(\varepsilon_s; \psi_\phi(\xi, y), \xi, y)$, $\varepsilon_s \sim \mathcal{N}(0, I)$. Here, $\psi_\phi(\xi, y)$ denotes the vector of variational parameters output by the amortizer that parameterizes $q_\phi(\boldsymbol{\theta} \mid \xi, y)$. This yields pathwise derivatives $\partial \boldsymbol{\theta}_s(\xi, y)/\partial \xi$, which we propagate through the decision layer (see Section 3.2).

**Computational benefit.** Although our experiments focus on single-step design over a relatively small discrete time grid, each candidate design is evaluated by solving an additional robust-control optimization problem that depends on the posterior over parameters, and we optimize the design using gradients with respect to $\xi$. Amortized variational inference lets us learn a single network that maps $(\xi, y)$ to approximate posterior parameters once, and then reuse this amortizer during the design phase to obtain both the posterior parameters and their design sensitivities (via the Jacobians $\partial \boldsymbol{\theta}/\partial \xi$ computed by AD). This removes the need to re-solve a full posterior inference problem for every candidate $\xi$ and every gradient step. As a result, the end-to-end computational cost is substantially reduced, especially when performing gradient-based search over a design space and repeatedly backpropagating through the decision layer.

## 4 RESULTS

We present numerical experiments on robust decision-making for goal-driven experimental design for three use cases, in order to demonstrate the efficacy and general applicability of the proposed GoBOED framework: an intuitive *2D source-location* problem, and two dynamical systems under model uncertainty, *epidemic management (SIQR)* and *pharmacokinetic (PK) control*. Detailed model parameters and solver settings are provided in Appendix F.

**Source localization with a single sensor.** As an intuitive example, we consider a two-dimensional source-location problem in which a single sensor must be placed to localize an emitting source and support a downstream intervention decision. The unknown parameter $\boldsymbol{\theta}$ encodes the coordinates of a point source, and the design variable is the sensor location $\xi \in \mathbb{R}^2$. Following Foster et al. (2021), we augment a scalar intensity measurement with a coarse angular observation (encoded via $\cos$ and $\sin$ of a bearing), which makes the posterior over $\boldsymbol{\theta}$ essentially unimodal and facilitates stable amortized inference.

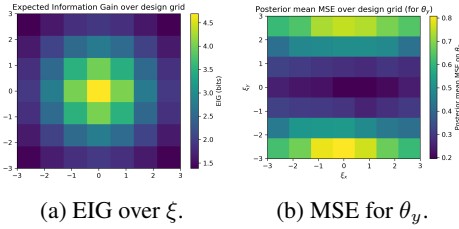

(a) EIG over $\xi$.  (b) MSE for $\theta_y$.

Figure 2: Source-location toy problem on a $7 \times 7$ grid of sensor locations $\xi$. (a) EIG is maximized near the center of the domain. (b) The decision-focused metric (posterior mean MSE of the source $y$-coordinate) is minimized along a horizontal band, yielding a broader near-optimal region for GoBOED than for standard EIG-based design.

On a $7 \times 7$ grid of candidate sensor locations, we compare standard BOED, which maximizes EIG about the source position, with GoBOED, which optimizes a decision-focused objective measuring the accuracy of the recovered source location. For EIG, larger values indicate more informative designs, whereas for the MSE objective, smaller values indicate better performance. For clarity we treat the vertical source coordinate $\theta_y$ as the decision-relevant quantity: the downstream decision reduces to estimating $\theta_y$ as accurately as possible, so that effectively only one unknown parameter matters for the objective. Both the EIG and MSE are estimated via nested Monte Carlo, using 10,000 outer samples and 10,000 inner samples per candidate design. Figure 2a shows the EIG surface over sensor locations, while Figure 2b displays the corresponding posterior mean squared error for $\theta_y$. The EIG-optimal design lies near the center of the domain, whereas the decision-focused objective exhibits a broad valley of near-optimal designs along a horizontal band, illustrating that purely information-driven designs need not align with robust localization performance even in this simple setting.

This example mirrors the qualitative behavior observed in our larger case studies, but with a single scalar decision and a simple loss. We now turn to SIQR and PK models, where the downstream decision is a higher-dimensional optimal control policy obtained by solving a convex program at each candidate design, and where the control actions influence the nonlinear system dynamics in a complex way.

To compare GoBOED with standard BOED baselines in these settings, we study the problem of choosing a single observation time. Let $T$ denote the time horizon and let $\boldsymbol{\xi} \in \{0, 1\}^T$ be a one-hot design vector with $\sum_{t=1}^{T} \xi_t = 1$; the unique index $t^*$ with $\xi_{t^*} = 1$ is the chosen measurement time. We collect exactly one measurement at $t^*$ and use this datum to update the posterior over model parameters. Because both the SIQR and PK settings are time-indexed, this formulation applies to both. At $t^*$, we observe the counts of asymptomatic and symptomatic infections for the SIQR model and the drug concentration in blood for the PK model; measurement noise is modeled as Poisson or Gaussian, respectively, depending on the data modality.

BOED via EIG targets maximal reduction in parameter uncertainty, whereas the robust decision-making objective is sensitive to particular parameter combinations and system dynamics. Accordingly, for each candidate design $\boldsymbol{\xi}$ we compute and visualize both the EIG and the robust optimal control cost over the entire design space. The resulting design objective surfaces for the SIQR (top) and PK models (bottom) are shown in Figure 3, with higher EIG and lower control cost indicating better designs.

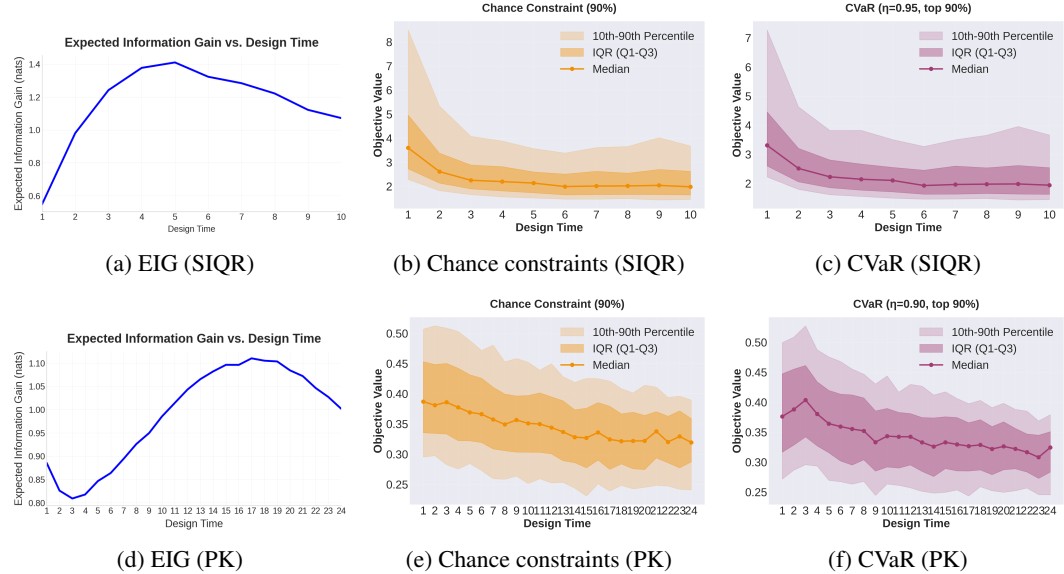

Figure 3: Comparison of experimental design metrics and control strategies across two models. Top row: SIQR epidemiological model—(a) EIG over observation time $\xi$; (b) expected optimal cost under chance constraints; (c) expected optimal cost under CVaR. Bottom row: pharmacokinetic (PK) model—(d) EIG; (e) chance constraints; (f) CVaR.

The horizontal axis is observation time $\xi$; The vertical axis shows EIG or expected optimal control costs. While the BOED-optimal design typically pinpoints a specific time, the goal-driven robust objective admits a broader near-optimal window—offering greater scheduling flexibility under real-world constraints.

We estimate EIG (cf. eq. (1)) using nested Monte Carlo with $5,000$ outer samples (over $y$) and $3,000$ inner samples for the marginal likelihood. The BOED-selected optimal observation times are day 5 for the SIQR model and hour 17 for the PK model.

**Robust optimal control in the presence of model uncertainty.** We solve the chance-constrained problem in eq. (9), enforcing a $90\%$ probability of constraint satisfaction under the posterior induced by a given observation time. Empirically, designs with larger EIG reduce constraint uncertainty and thereby yield lower optimal cost in both examples. For SIQR, the objective is relatively flat for observation times between days 4 and 8; for PK, observing later (roughly 15–23 hours) reduces the dose required to meet the therapeutic targets. Unless otherwise noted, we draw $500$ datasets $y$ and, for each $y$, $40$ posterior samples of $\boldsymbol{\theta}$ (128 for PK).

**CVaR-based constraints.** We also consider the CVaR formulation in Section 3.1.2: for SIQR we constrain the CVaR of the dominant eigenvalue, and for PK we constrain the CVaR of $C_{\max}$ relative to its threshold. We set the level to $\alpha = 0.9$ (controlling the expected violation over the worst $10\%$ of posterior realizations). The qualitative trends mirror those under chance constraints: higher-EIG designs generally achieve lower robust cost; SIQR exhibits a plateau around days 4–8, and in PK a later observation (e.g., near 24 hours) further reduces the dose required to achieve the target performance. We use the same sampling budgets as above ($500$ draws of $y$; $40$ posterior draws per $y$, or 128 for PK).

These results highlight a key insight: maximizing EIG alone does not necessarily optimize constraint-aware economic performance. Instead, our approach identifies a broad near-optimal window for SIQR—approximately days 4–8—within which observation timing can be chosen with minimal loss in both information gain and economic efficiency. For PK dose optimization, we similarly find a wide near-optimal window (about 15–23 hours). Within this window, latter observations (toward 22–23 hours) generally yield robust control with lower cost than the BOED-optimal 17 hours while achieving comparable EIG. This scheduling flexibility is practically valuable when exact timing for observation measurements is restricted due to logistical or operational constraints.

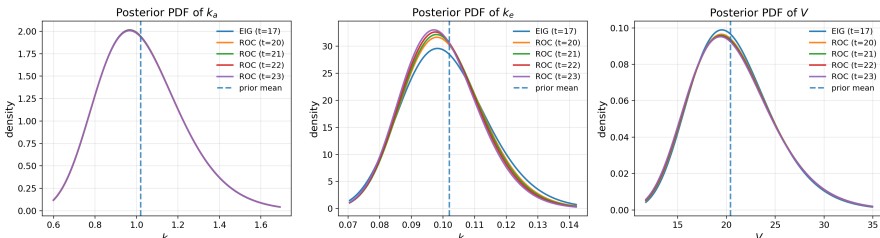

Figure 4: Posterior densities for the PK parameters $k_a$, $k_e$, and $V$ under different optimal designs.

**Posterior visualization.** We now compare the posterior distributions obtained under EIG-based and decision-focused designs. Figure 4 displays the marginal posteriors for the PK parameters $k_a$, $k_e$, and $V$. The blue curves correspond to the EIG-optimal design at $t = 17$, and the orange and green curves correspond to GoBOED designs at $t = 20, 21, 22, 23$, respectively. The dashed vertical line indicates the prior mean used as the data-generating parameter in the forward model; the resulting simulated data are then used to estimate the posterior distributions. All five designs produce similar posterior uncertainty for $k_a$ and $V$, whereas the GoBOED designs yield a sharper and shifted posterior for $k_e$, reflecting the fact that GoBOED targets parameter combinations that are most influential for the downstream control objective. We can check that GoBOED also concentrates posterior mass on decision-relevant parameter combinations in the SIQR model as shown in Appendix F.3, Figure 5.

**Gradient-based search over discrete times.** We treat the observation time as a scalar $t \in \{1, \ldots, T\}$ (equivalently, a one-hot $\boldsymbol{\xi}(t)$) and directly optimize $t$ using automatic differentiation. Starting from a mid-point integer $t_0 = \lfloor(T + 1)/2\rfloor$, we compute the AD gradient of the robust objective with respect to $t$ and take projected gradient steps:

$$s_k = \left.\frac{\partial J(t)}{\partial t}\right|_{t=t_k}, \qquad t_{k+1} = \Pi_{[1,T]}\big(t_k - \omega_k\, s_k\big), \qquad \hat{t}_{k+1} = \text{round}(t_{k+1}),$$

where $\Pi_{[1,T]}$ clips to the feasible interval and $\omega_k$ is the step size. After each update, we evaluate the design at the integer index $\hat{t}_{k+1}$ by recomputing the posterior induced by a single observation at $\hat{t}_{k+1}$ and then re-evaluating the robust optimal cost $J(\hat{t}_{k+1})$. We repeat these steps until convergence.

For the SIQR model, the proposed procedure converges near the BOED design by day 5, with days 6–7 showing similarly small gradient norms. For the PK model, it selects the 22-hour design for both the CVaR and chance-constrained criteria, which aligned with the plot in Figure 3. We also confirmed a monotonic decrease in the gradient norm, consistent with the trend observed in the plot.

## 5 CONCLUSION

We developed a computational methodology that jointly trains a posterior approximation and optimizes uncertainty-aware, goal-driven experimental decisions, thereby enabling robust optimal management and control under model uncertainty. By bridging convex optimal control with BOED, the framework supports real-world deployments that could meaningfully influence epidemic response strategies and dosing optimization while improving computational efficiency. Applying the method to large-scale epidemiology, clinical, and drug discovery datasets—and integrating it into production workflows—remains important future work.

**Limitations.** Our framework assumes convex optimization, which enables differentiation through the control layer via Lagrangian sensitivity analysis. This assumption is both a strength and a constraint: for non-convex objectives we typically obtain only locally optimal solutions, increasing optimization difficulty. Performance also depends on the quality of variational inference; inaccuracies in the learned posterior can degrade decisions and undermine reliability, particularly in real-world settings. Future work should strengthen robustness and expand posterior expressivity—for example, by leveraging generative families such as diffusion and flow-based models. Extending GoBOED to fully sequential designs is another important direction. Doing so would require architectures that efficiently update or condition on posteriors after each measurement (e.g., in the spirit of Huang et al. (2024) or recursive frameworks such as Foster et al. (2021); Ivanova et al. (2021; 2024)).

**LLM Usage:** This manuscript was copy-edited for grammar and style using ChatGPT (OpenAI; accessed September 2025). The authors drafted all text and iteratively reviewed and revised AI-suggested edits. All ideas, methods, analyses, and conclusions were developed solely by the authors, who accept full responsibility for the final content. No confidential or reviewer-only material was provided to the model.

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

## A    NOTATIONS

Table 1 provides the detailed list of mathematical notations and symbols adopted in GoBOED.

| Symbol | Description |
|---|---|
| **Bayesian model and experimental design** | |
| $\boldsymbol{\theta} \in \Theta$ | Unknown model parameters (e.g., SIQR or PK parameters) |
| $\xi \in \Xi$ | Experimental design (e.g., observation time) |
| $y \in \mathcal{Y}$ | Observation generated under design $\xi$ |
| $p(\boldsymbol{\theta})$ | Prior over model parameters |
| $p(y \mid \boldsymbol{\theta}, \xi)$ | Likelihood (forward model plus observation noise) |
| $p(y \mid \xi)$ | Marginal likelihood under design $\xi$ |
| $f(\boldsymbol{\theta}, \xi)$ | Forward model mapping $(\boldsymbol{\theta}, \xi)$ to noiseless output |
| $\epsilon$ | Observation noise term |
| $\mathrm{EIG}(\xi)$ | Expected information gain objective for design $\xi$ |
| $D_{\mathrm{KL}}(\cdot \| \cdot)$ | Kullback–Leibler divergence |
| **Variational inference and amortizer** | |
| $q_\phi(\boldsymbol{\theta} \mid \xi, y)$ | Variational approximation to $p(\boldsymbol{\theta} \mid \xi, y)$ with parameters $\phi$ |
| $\phi$ | Parameters of the amortized VI network |
| $\psi_\phi(\xi, y)$ | Variational parameters output by the amortizer |
| $\mathcal{L}_{\mathrm{VI}}(\phi; y, \xi)$ | Evidence lower bound (ELBO) for $(y, \xi)$ |
| $h_\phi(\varepsilon; \psi_\phi(\xi, y), \xi, y)$ | Reparameterization map turning base noise $\varepsilon$ into $\boldsymbol{\theta}$ samples |
| $w_i, \tilde{w}_i$ | Importance weights and their normalized versions for sample $i$ |
| **Robust control and risk measures** | |
| $\boldsymbol{g} \in \mathcal{G}$ | Decision / control variables (e.g., quarantine rates or dose) |
| $J(\boldsymbol{g}; \boldsymbol{\theta})$ | Application cost for decision $\boldsymbol{g}$ and parameters $\boldsymbol{\theta}$ |
| $\rho(\cdot)$ | Risk functional (expectation, chance constraint, or CVaR) |
| $\eta$ | Confidence / risk level for chance constraints or CVaR |
| $v_i(\boldsymbol{g}; \boldsymbol{\theta}_i)$ | Constraint violation for posterior sample $\boldsymbol{\theta}_i$ |
| $\tau$ | Auxiliary threshold variable in CVaR formulation |
| $s_i$ | Slack variable for CVaR constraint for sample $i$ |
| $\lambda_i$ | Lagrange multiplier for constraint of sample $i$ (decision layer) |
| **Optimization and design objective** | |
| $L(\xi, \phi)$ | Design loss combining robust control and VI terms |
| $\omega_\xi, \omega_\phi$ | Step sizes for updating $\xi$ and $\phi$ in gradient-based optimization |
| $t$ | Discrete observation time index (when $\xi$ encodes timing) |

Table 1: Summary of notations used for Bayesian optimal experimental design (BOED), variational inference, and robust decision-making in GoBOED.

## B    ALGORITHMIC AND INFERENCE DETAILS

### B.1    GoBOED ALGORITHM (PSEUDOCODE)

Algorithm 1 summarizes the two-stage GoBOED procedure: offline training of the amortized variational posterior, followed by gradient-based optimization of the design $\xi$ under the robust decision-making objective.

### B.2    VARIATIONAL NETWORK ARCHITECTURE

Let $\xi \in \Xi$ denote the design and $y \in \mathcal{Y}$ the observation. Both are mapped into a shared latent space using linear tokenizers that produce $M$ tokens of width $d$:

$$E_\xi : \mathbb{R}^{D_\xi} \to \mathbb{R}^{M \times d}, \quad E_y : \mathbb{R}^{D_y} \to \mathbb{R}^{M \times d}, \qquad Z_\xi = E_\xi(\xi), \ Z_y = E_y(y),$$

where in PK we set $D_\xi = 1$, $D_y = 1$, $M = 8$, and $d = 64$. Queries and keys are linear projections of the token sequences,

$$q_j = W_q \, z_{y,j} \in \mathbb{R}^d, \qquad k_i = W_k \, z_{\xi,i} \in \mathbb{R}^d,$$

---

**Algorithm 1** GoBOED: Goal-driven Bayesian Optimal Experimental Design

---

**Require:** Prior $p(\boldsymbol{\theta})$, design space $\Xi$, forward model $p(y \mid \boldsymbol{\theta}, \xi)$, robust control problem (cost $J(\boldsymbol{g}; \boldsymbol{\theta})$, constraints), amortized VI network $q_\phi(\boldsymbol{\theta} \mid y, \xi)$, step sizes $\omega_\phi, \omega_\xi$, Monte Carlo batch size $M$ for design optimization

**Ensure:** Approximate optimal design $\xi^*$ and variational parameters $\phi^*$

    **Offline: train amortized variational inference**
1: Initialize variational parameters $\phi$
2: **while** not converged **do**
3:     Sample $\xi \sim p(\xi)$, $\boldsymbol{\theta} \sim p(\boldsymbol{\theta})$, $y \sim p(y \mid \boldsymbol{\theta}, \xi)$
4:     Compute ELBO $\mathcal{L}_{\mathrm{VI}}(\phi; y, \xi)$ as in equation 6
5:     Update $\phi \leftarrow \phi + \omega_\phi \nabla_\phi \mathcal{L}_{\mathrm{VI}}(\phi; y, \xi)$
6: **end while**
    **Design optimization: decision-focused loop**
7: Define design loss $L(\xi) = \mathbb{E}_{p(y|\xi)}\big[\mathcal{L}_{\mathrm{ROC}}(y, \xi; \phi)\big]$
8: Initialize design $\xi$ (e.g. mid-point of allowed time window)
9: **while** not converged **do**
10:     Set Monte Carlo estimate $\hat{L}(\xi) \leftarrow 0$
11:     **for** $m = 1, \ldots, M$ **do**
12:         Sample $y^{(m)} \sim p(y \mid \xi)$
13:         Build posterior $q_\phi(\boldsymbol{\theta} \mid y^{(m)}, \xi)$ and draw $\boldsymbol{\theta}$ via reparameterization
14:         Solve the robust decision-making problem under $q_\phi(\boldsymbol{\theta} \mid y^{(m)}, \xi)$ via the decision layer, obtaining $\mathcal{L}_{\mathrm{ROC}}(\boldsymbol{\theta}; y^{(m)}, \xi, \phi)$
15:         $\hat{L}(\xi) \leftarrow \hat{L}(\xi) + \mathcal{L}_{\mathrm{ROC}}(\boldsymbol{\theta}; y^{(m)}, \xi, \phi)$
16:     **end for**
17:     $\hat{L}(\xi) \leftarrow \hat{L}(\xi)/M$
18:     Compute stochastic gradient $g_\xi \leftarrow \nabla_\xi \hat{L}(\xi)$ using reparameterization + implicit differentiation
19:     Update $\xi \leftarrow \xi - \omega_\xi g_\xi$
20: **end while**
21: **return** $\xi^* \leftarrow \xi$, $\phi^* \leftarrow \phi$

---

and values fuse per-token key and query latents via a small MLP,

$$v_i = \Psi\big([\, z_{\xi,i},\, z_{y,i}\,]\big) \in \mathbb{R}^d, \qquad i, j \in \{1, \ldots, M\}.$$

Single-head dot-product cross-attention over tokens is

$$a_{ji} = \frac{\exp\big(q_j^\top k_i / \sqrt{d}\big)}{\sum_{i'=1}^{M} \exp\big(q_j^\top k_{i'} / \sqrt{d}\big)}, \qquad c_j = \sum_{i=1}^{M} a_{ji}\, v_i \in \mathbb{R}^d.$$

We mean-pool the query contexts and pass through a light trunk MLP:

$$s = \frac{1}{M} \sum_{j=1}^{M} c_j, \qquad h = \mathrm{MLP}_{\mathrm{trunk}}(s) \in \mathbb{R}^d.$$

Two heads with skip connections produce log-space parameters for a diagonal LogNormal posterior,

$$\tilde{\mu} = W_{\ell,2}\, \phi(W_{\ell,1} h) + W_{\ell,\mathrm{skip}} h,$$
$$\tilde{\sigma} = W_{s,2}\, \phi(W_{s,1} h) + W_{s,\mathrm{skip}} h,$$

which we bound elementwise:

$$\mu = \mu_0 + \Delta_{\max} \tanh(\tilde{\mu}), \qquad \sigma = \sigma_{\min} + (\sigma_{\max} - \sigma_{\min})\, \boldsymbol{\sigma}(\tilde{\sigma}).$$

Here $\phi$ is GELU, $\mu_0$ is the prior log-mean (used as a residual center), $\Delta_{\max}$ bounds deviations, $0 < \sigma_{\min} < \sigma_{\max}$ bound the log-space standard deviations, and $\boldsymbol{\sigma}$ is sigmoid activation function.

### B.3 OPTIMIZATION AND SOLVERS

We model the semidefinite programs arising in the decision layer and solve them using SCS O'Donoghue et al. (2023) with default settings. For comparison and smaller instances, we also solved the problems with the interior-point solver MOSEK ApS (2025) and obtained numerically indistinguishable solutions.

### B.4 IMPLEMENTATION AND TRAINING CONFIGURATION

We train all amortized variational posteriors $q_\phi(\cdot \mid y, \xi)$ offline, following Algorithm 1. For each experiment we sample design–observation pairs $(\xi, y)$ from the prior predictive model and optimize the ELBO in eq. (6) using stochastic gradient descent.

We use the Adamw optimizer Loshchilov & Hutter (2017) with learning rate $10^{-4}$ and train for $500$ epochs on $1,000$ $(\xi, y)$ pairs per experiment. The designs $\xi$ are chosen on a fixed grid over the design space ($7 \times 7$ grid points for source localization, 20 points for the SIQR experiment, and 24 points for the PK experiment). For the inner expectations in the ELBO we draw $400$ posterior samples from $q_\phi(\cdot \mid y, \xi)$ per $(\xi, y)$ pair. Gradients are computed via automatic differentiation through the forward models and the decision layer. All experiments are run on a single NVIDIA A100 GPU.

## C   MODEL AND CONTROL FORMULATIONS

### C.1   SOURCE LOCALIZATION TOY MODEL

We consider a two-dimensional source-localization problem in which a single sensor must be placed to localize an emitting point source. The unknown parameter $\boldsymbol{\theta} \in \mathbb{R}^2$ encodes the Cartesian coordinates of the source, written as $\boldsymbol{\theta} = (\theta_x, \theta_y)^\top$. We place an independent standard normal prior on each coordinate,

$$\boldsymbol{\theta} \sim \mathcal{N}(0, I_2).$$

The design variable is the sensor location $\xi \in \mathbb{R}^2$. For a given design $\xi$ and parameter $\boldsymbol{\theta}$, the forward map returns a three-dimensional summary of the local signal field. Let

$$d(\xi, \boldsymbol{\theta}) = \boldsymbol{\theta} - \xi \in \mathbb{R}^2, \qquad r^2(\xi, \boldsymbol{\theta}) = \|d(\xi, \boldsymbol{\theta})\|_2^2$$

denote the offset and squared distance from the sensor to the source. We assign an inverse-square-type weight

$$w(\xi, \boldsymbol{\theta}) = \frac{1}{s_0 + r^2(\xi, \boldsymbol{\theta})},$$

with $s_0 > 0$ a small regularization constant, and define the total intensity

$$I(\xi, \boldsymbol{\theta}) = c_{\text{base}} + w(\xi, \boldsymbol{\theta}),$$

where $c_{\text{base}}$ is a background level.

We also summarize the bearing from the sensor towards the source. Since there is only one source, this is simply the direction of $d(\xi, \boldsymbol{\theta})$. Let

$$\bar{\psi}(\xi, \boldsymbol{\theta}) = \text{atan2}\big(d_y(\xi, \boldsymbol{\theta}), d_x(\xi, \boldsymbol{\theta})\big)$$

denote the polar angle of the offset. To avoid angular discontinuities we encode this angle via $(\cos \bar{\psi}, \sin \bar{\psi})$. The noiseless observation is therefore

$$m(\xi, \boldsymbol{\theta}) = \big(\log I(\xi, \boldsymbol{\theta}),\ \cos \bar{\psi}(\xi, \boldsymbol{\theta}),\ \sin \bar{\psi}(\xi, \boldsymbol{\theta})\big) \in \mathbb{R}^3,$$

and we model the noisy measurement as

$$y \mid \boldsymbol{\theta}, \xi \sim \mathcal{N}\big(m(\xi, \boldsymbol{\theta}),\ \text{diag}(\sigma_I^2, \sigma_\phi^2, \sigma_\phi^2)\big),$$

with different noise scales for intensity and angular components.

For this toy problem we treat the vertical source coordinate $\theta_y$ as the decision-relevant quantity. Given an observation $(\xi, y)$, the downstream "decision" is the scalar estimate $\hat{\theta}_y(\xi, y)$ produced by the amortized posterior network $q_\phi(\boldsymbol{\theta} \mid \xi, y)$. We use the squared error in $\theta_y$ as the loss,

$$\text{MSE}(\xi, y; \boldsymbol{\theta}) = \big(\hat{\theta}_y(\xi, y) - \theta_y\big)^2,$$

and the decision-focused design objective is the expected posterior mean squared error

$$J(\xi) = \mathbb{E}_{\boldsymbol{\theta},y}\big[\text{MSE}(\xi, y; \boldsymbol{\theta})\big],$$

where the expectation is taken over the prior $p(\boldsymbol{\theta})$ and the likelihood $p(y \mid \boldsymbol{\theta}, \xi)$.

## C.2 OPTIMAL CONTROL FOR EPIDEMIOLOGY

Building on the BOED framework, we consider robust epidemiology control as an example. Our method leverages the framework established by Talaei et al. (2024). The epidemiology model is governed by a SIQR spread disease network, where state variables represent different compartments of the population: susceptible ($s$), asymptomatic infected ($x^a$), symptomatic infected ($x^s$) and recovered ($h$).

$$\begin{pmatrix} \dot{s} \\ \dot{x}^a \\ \dot{x}^s \\ \dot{h} \end{pmatrix} = \begin{pmatrix} 0 & -\beta^a s & -\beta^s s & 0 \\ 0 & \beta^a s - \epsilon - \gamma^a - g^a & \beta^s s & 0 \\ 0 & \epsilon & -\gamma^s - g^s & 0 \\ 0 & \gamma^a & \gamma^s & 0 \end{pmatrix} \begin{pmatrix} s \\ x^a \\ x^s \\ h \end{pmatrix}. \tag{13}$$

Here, $\beta^a$ and $\beta^s$ are transmission rates for asymptomatic and symptomatic cases, $\epsilon$ is the rate at which asymptomatic cases develop symptoms, $\gamma^a$ and $\gamma^s$ are recovery rates for asymptomatic and symptomatic cases, and $g^a$ and $g^s$ are quarantine rates for asymptomatic and symptomatic individuals.

Following Ma et al. (2023), we decouple the dynamics of $\dot{x}$ from $\dot{s}$ and $\dot{h}$, allowing us to focus on the matrix $M(t_0)$ which captures the essential infection dynamics at initial time $t_0$:

$$M(t_0) = \begin{pmatrix} \beta^a s(t_0) - \epsilon - \gamma^a - g^a & \beta^s s(t_0) \\ \epsilon & -\gamma^s - g^s \end{pmatrix} \tag{14}$$

To optimize the quarantine strategy as described in Talaei et al. (2024), we utilize an objective function that minimizes economic costs:

$$\min_{g^a, g^s} J(g^a, g^s) = \frac{z^a}{1 - g^a} + \frac{z^s}{1 - g^s} \tag{15}$$

$$\text{s.t. } \lambda_{\max}(M(t_0)) \leq -\alpha, \tag{16}$$

$$0 \leq g^a < 1, \tag{17}$$

$$0 \leq g^s < 1, \tag{18}$$

where $z^a$ represents the economic cost for asymptomatic quarantine, $z^s$ represents the economic cost for symptomatic quarantine, and $\alpha > 0$ is a constraint ensuring the stability of the system.

The detailed solution for this minimization problem is provided in Appendix D. This approach allows us to determine optimal quarantine rates without explicitly integrating the SIQR differential equations. Instead, by analyzing the eigenvalues of the system and applying convex optimization, we can efficiently identify the optimal quarantine strategy that minimizes economic costs.

## C.3 OPTIMAL CONTROL FOR PHARMACOKINETIC MODEL

We further consider PK model as another example. The concentration at time $t$ for a drug administered orally can be modeled using the Bateman function (Bateman, 1910):

$$y(t) = \frac{D \cdot k_a}{V \cdot (k_a - k_e)} \left( e^{-k_e \cdot t} - e^{-k_a \cdot t} \right) (1 + \epsilon_{\text{mult}}) + \epsilon_{\text{add}},$$

where $V$ is the volume of distribution, $k_a$ is the absorption rate constant, $k_e$ is the elimination rate constant, $D$ is the dose administered, $\epsilon_{\text{mult}}$ is a multiplicative error term, and $\epsilon_{\text{add}}$ is additive error term. This formulation captures the dynamics of drug absorption and elimination.

Given the drug's potential toxicity, dosing should maintain systemic exposure within the therapeutic window—avoiding toxic concentrations while not falling below the minimum effective concentration. We can calculate the time at which the maximum drug concentration occurs, denoted $t_{\max}$, can be found by setting the derivative $\partial y / \partial t = 0$, which yields:

$$t_{\max} = \frac{\ln(k_a/k_e)}{k_a - k_e}.$$

The maximum concentration, $C_{\max}$, is obtained by evaluating $y(t)$ at $t_{\max}$:

$$C_{\max} = \frac{D}{V} \left( \frac{k_e}{k_a} \right)^{\frac{k_e}{-k_e+k_a}} (1 + \epsilon_{\text{mult}}) + \epsilon_{\text{add}}.$$

For cumulative exposure, the area under the concentration curve (AUC) is given by:

$$\text{AUC} = \int_0^\infty y(t)dt = \frac{D}{V \cdot k_e},$$

assuming complete absorption.

We define a convex cost function $J(g)$ (e.g., $J(g) = c\,g$ to discourage high dosing). The constrained problem can be written in the following forms,

$$\begin{aligned}
\min_{0 \le g \le 1} \quad & J(g) \qquad\qquad\qquad\qquad\qquad\qquad (19)\\
\text{s.t.} \quad & C_{\max}(g, \boldsymbol{\theta}) \le C_{\text{thresh}},\\
& \text{AUC}(g, \boldsymbol{\theta}) \ge \text{AUC}_{\min}.
\end{aligned}$$

## D  QUARANTINE OPTIMIZATION AND LAGRANGIAN FORMULATION

We now derive gradients for the SIQR control problem in eq. (15) using a Lagrangian and KKT conditions. Here $(\boldsymbol{\beta}_i, \boldsymbol{\gamma}_i)$ denote the sampled transmission and recovery rates for scenario $i$, as introduced in Appendix C.2.

To address this constrained optimization problem, we introduce the Lagrangian:

$$\begin{aligned}
\mathcal{L} = \frac{z^a}{1-g^a} + \frac{z^s}{1-g^s} + \sum_{i=1}^N \lambda_i (\lambda_{\max}(M(t_0, \boldsymbol{\beta}_i, \boldsymbol{\gamma}_i, g^a, g^s)) + \alpha) \\
- \mu_1 g^a + \mu_2(g^a - 1) - \mu_3 g^s + \mu_4(g^s - 1),
\end{aligned}$$

where $\lambda_i$ are Lagrange multipliers associated with the eigenvalue constraints, and $\mu_i$ are multipliers for the box constraints on $g^a$ and $g^s$. The optimal solution must satisfy the Karush-Kuhn-Tucker (KKT) conditions, which we outline below.

### D.1  KKT CONDITIONS

**Stationarity**  The stationarity conditions are derived by taking partial derivatives of the Lagrangian:

$$\frac{\partial \mathcal{L}}{\partial g^a} = \frac{z^a}{(1-g^a)^2} + \sum_{i=1}^N \lambda_i \frac{\partial \lambda_{\max}(M(t_0, \boldsymbol{\beta}_i, \boldsymbol{\gamma}_i, g^a, g^s))}{\partial g^a} + \mu_2 - \mu_1 = 0,$$

$$\frac{\partial \mathcal{L}}{\partial g^s} = \frac{z^s}{(1-g^s)^2} + \sum_{i=1}^N \lambda_i \frac{\partial \lambda_{\max}(M(t_0, \boldsymbol{\beta}_i, \boldsymbol{\gamma}_i, g^a, g^s))}{\partial g^s} + \mu_4 - \mu_3 = 0.$$

**Primal feasibility**  The primal feasibility conditions ensure the constraints hold:

$$\begin{aligned}
\lambda_{\max}(M(t_0, \boldsymbol{\beta}_i, \boldsymbol{\gamma}_i, g^a, g^s)) &\le -\alpha \quad \text{for} \quad i = 1, \ldots, N,\\
0 &\le g^a < 1,\\
0 &\le g^s < 1.
\end{aligned}$$

**Dual feasibility**  The Lagrange multipliers must be non-negative:

$$\lambda_i \ge 0 \quad \text{for} \quad i = 1, \ldots, N, \quad \text{and} \quad \mu_1, \mu_2, \mu_3, \mu_4 \ge 0.$$

**Complementary slackness**   The complementary slackness conditions are:

$$\lambda_i \left( \lambda_{\max}(M(t_0, \boldsymbol{\beta}_i, \boldsymbol{\gamma}_i, g^a, g^s)) + \alpha \right) = 0 \quad \text{for} \quad i = 1, \ldots, N,$$
$$\mu_1 g^a = 0,$$
$$\mu_2(1 - g^a) = 0,$$
$$\mu_3 g^s = 0,$$
$$\mu_4(1 - g^s) = 0.$$

Assuming an interior solution, the multipliers for the box constraints at the optimal point become $\mu_i = 0$. The KKT conditions simplify to:

$$\frac{z^a}{(1 - g^{a*})^2} + \sum_{i=1}^{N} \hat{\lambda}_i \frac{\partial \lambda_{\max}(M(t_0, \boldsymbol{\beta}_i, \boldsymbol{\gamma}_i, g^{a*}, g^{s*}))}{\partial g^a} = 0,$$

$$\frac{z^s}{(1 - g^{s*})^2} + \sum_{i=1}^{N} \hat{\lambda}_i \frac{\partial \lambda_{\max}(M(t_0, \boldsymbol{\beta}_i, \boldsymbol{\gamma}_i, g^{a*}, g^{s*}))}{\partial g^s} = 0,$$

$$\lambda_{\max}(M(t_0, \boldsymbol{\beta}_{i^*}, \boldsymbol{\gamma}_{i^*}, g^{a*}, g^{s*})) = -\alpha \quad \text{for specific} \quad i^*,$$

where $\hat{\lambda}_i$ denotes the optimal Lagrange multipliers.

We compute the optimal values $g^{a*}$, $g^{s*}$, and $\hat{\lambda}_{i^*}$ using convex optimization via semidefinite programming. This is feasible because $J$ is a convex function with respect to $g^a$ and $g^s$, and the largest eigenvalue constraint can be reformulated as a set of linear matrix inequalities.

## D.2   DERIVATIVE OF THE OPTIMUM COST $J^*$ WITH RESPECT TO MODEL PARAMETERS

We can compute the gradient of the optimum cost $J^*$ with respect to the model parameters $\boldsymbol{\beta}$ and $\boldsymbol{\gamma}$. Using the envelope theorem, the derivative with respect to $\beta_i^a$ is:

$$\frac{\partial J^*}{\partial \beta_i^a} = \hat{\lambda}_i \cdot \frac{\partial \lambda_{\max}\left(M(t_0, \boldsymbol{\beta}_i, \boldsymbol{\gamma}_i, g^{a*}, g^{s*})\right)}{\partial \beta_i^a}.$$

Similarly, the derivatives with respect to other parameters are:

$$\frac{\partial J^*}{\partial \gamma_i^s} = \hat{\lambda}_i \cdot \frac{\partial \lambda_{\max}(M(t_0, \boldsymbol{\beta}_i, \boldsymbol{\gamma}_i, g^{a*}, g^{s*}))}{\partial \gamma_i^s},$$

$$\frac{\partial J^*}{\partial \gamma_i^a} = \hat{\lambda}_i \cdot \frac{\partial \lambda_{\max}(M(t_0, \boldsymbol{\beta}_i, \boldsymbol{\gamma}_i, g^{a*}, g^{s*}))}{\partial \gamma_i^a},$$

$$\frac{\partial J^*}{\partial \beta_i^s} = \hat{\lambda}_i \cdot \frac{\partial \lambda_{\max}(M(t_0, \boldsymbol{\beta}_i, \boldsymbol{\gamma}_i, g^{a*}, g^{s*}))}{\partial \beta_i^s}.$$

Since the optimal solution often lies on specific boundaries where $\hat{\lambda}_{i^*} \neq 0$ and $\hat{\lambda}_i = 0$ for $i \neq i^*$, the gradient depends only on the samples directly influencing the solution.

## D.3   DERIVATIVE OF THE CONSTRAINTS WITH RESPECT TO EXPERIMENTAL DESIGN

The constraint term is defined as:

$$l(\boldsymbol{\beta}, \boldsymbol{\gamma}, g^a, g^s; y, \xi) = \sum_{i=1}^{N} \lambda_i \left( \lambda_{\max}(M(t_0, \boldsymbol{\beta}_i, \boldsymbol{\gamma}_i, g^a, g^s)) + \alpha \right) - \mu_1 g^a + \mu_2(g^a - 1) - \mu_3 g^s + \mu_4(g^s - 1).$$

Its gradient with respect to $\xi$ at the optimal point is:

$$\nabla_\xi l = \sum_{i=1}^{N} \hat{\lambda}_i \left( \frac{\partial \lambda_{\max}}{\partial \beta_i^a} \cdot \nabla_\xi \beta_i^a + \frac{\partial \lambda_{\max}}{\partial \beta_i^s} \cdot \nabla_\xi \beta_i^s + \frac{\partial \lambda_{\max}}{\partial \gamma_i^a} \cdot \nabla_\xi \gamma_i^a + \frac{\partial \lambda_{\max}}{\partial \gamma_i^s} \cdot \nabla_\xi \gamma_i^s \right). \qquad (20)$$

The partial derivative $\frac{\partial \lambda_{\max}}{\partial \beta_i^a}$ is given by Talaei et al. (2024) as:

$$\frac{\partial \lambda_{\max}}{\partial \beta_i^a} = \frac{v_{\max}^T \left( \frac{\partial M(t_0, \boldsymbol{\beta}_i)}{\partial \beta_i^a} \right) u_{\max}}{v_{\max}^T u_{\max}},$$

where $v_{\max}$ and $u_{\max}$ are the left and right eigenvectors of the largest eigenvalue, respectively. The terms $\nabla_\xi \beta_i^a$, $\nabla_\xi \beta_i^s$, $\nabla_\xi \gamma_i^a$, and $\nabla_\xi \gamma_i^s$ are computed via automatic differentiation from the variational network.

### D.4 MARGINAL LIKELIHOOD GRADIENT

To compute the log-likelihood gradient with respect to $\xi$, we assume a Poisson observation model (suitable for count data) with rate parameter $\lambda = 0.95 \cdot y_{\text{true}}(\xi)$. The gradient is:

$$\frac{\partial}{\partial \xi} \log p(y_{\text{obs}} \mid \xi) = \left( \frac{y_{\text{obs}}}{0.95 \cdot y_{\text{true}}(\xi)} - 1 \right) \cdot 0.95 \cdot \frac{\partial y_{\text{true}}(\xi)}{\partial \xi}.$$

This expression helps quantify how changes in $\xi$ affect the likelihood of the observed data. The term $\frac{\partial y_{\text{true}}(\xi)}{\partial \xi}$ can be approximated using finite difference methods, such as the central difference method.

## E ROBUST DECISION-MAKING

### E.1 SIQR MODEL

We develop an integrated framework, GoBOED, for epidemic management by bringing together BOED (introduced in eq. (2)) and optimal control through convex optimization (described in eq. (3)). Our primary objective is to identify an experimental design $\xi^*$ that minimizes the expected controlled economic cost, where the expectation is taken over the posterior distributions of the parameters $\boldsymbol{\beta} = (\beta^a, \beta^s)$ and $\boldsymbol{\gamma} = (\gamma^a, \gamma^s)$. This allows us to update our beliefs with new observations and improve decision-making, following the Bayesian decision-theoretic perspective of Chaloner & Verdinelli (1995).

For a given design $\xi$ and observed data $y$, we consider

$$\min_{g^a, g^s} \mathbb{E}_{p(\boldsymbol{\beta}, \boldsymbol{\gamma} \mid y, \xi)} \big[ J(g^a, g^s) \big], \qquad (21)$$

subject to the posterior-averaged stability constraint

$$\lambda_{\max} \Big( \mathbb{E}_{p(\boldsymbol{\beta}, \boldsymbol{\gamma} \mid y, \xi)} \big[ M(t_0, \boldsymbol{\beta}, \boldsymbol{\gamma}, g^a, g^s) \big] \Big) \leq -\alpha, \qquad (22)$$
$$0 \leq g^a < 1, \quad 0 \leq g^s < 1.$$

Here, $J(g^a, g^s)$ denotes the economic cost, which does not depend explicitly on $(\boldsymbol{\beta}, \boldsymbol{\gamma})$; hence the objective in eq. (21) reduces to minimizing $J(g^a, g^s)$ subject to eq. (22). The main difficulty lies in evaluating the eigenvalue constraint, which involves the posterior expectation of the matrix $M(t_0, \boldsymbol{\beta}, \boldsymbol{\gamma}, g^a, g^s)$.

To approximate the posterior $p(\boldsymbol{\beta}, \boldsymbol{\gamma} \mid y, \xi)$ we employ the amortized variational inference scheme introduced in the main text. In particular, we use the variational family $q_\phi(\boldsymbol{\beta}, \boldsymbol{\gamma} \mid y, \xi)$ and optimize $\phi$ via the ELBO in eq. (6). Expectations under the true posterior are then estimated using the importance-sampling estimator defined in eq. (7). For the spectral stability constraint, we take $f(\boldsymbol{\beta}, \boldsymbol{\gamma}) = M(t_0, \boldsymbol{\beta}, \boldsymbol{\gamma}, g^a, g^s)$ in that estimator, which yields an approximation of $\mathbb{E}_{p(\boldsymbol{\beta}, \boldsymbol{\gamma} \mid y, \xi)}[M(t_0, \boldsymbol{\beta}, \boldsymbol{\gamma}, g^a, g^s)]$ and thereby of the eigenvalue constraint in eq. (22).

### E.1.1 ALTERNATIVE FORMULATION USING CHANCE CONSTRAINTS

To handle parameter uncertainty more explicitly, we also consider a chance-constrained formulation. Chance constraints ensure that a critical condition holds with a specified probability (Charnes & Cooper, 1959; Miller & Wagner, 1965). Here, the condition is spectral stability.

Rather than enforcing eigenvalue constraints for every posterior sample, which can be overly conservative, we require that the maximum eigenvalue of $M(t_0, \boldsymbol{\beta}, \boldsymbol{\gamma}, g^a, g^s)$ lies below $-\alpha$ with high posterior probability. The resulting problem is

$$\min_{g^a, g^s} J(g^a, g^s), \tag{23}$$

subject to

$$\mathbb{P}\big(\lambda_{\max}\big(M(t_0, \boldsymbol{\beta}, \boldsymbol{\gamma}, g^a, g^s)\big) \leq -\alpha \,\big|\, y, \xi\big) \geq \eta, \tag{24}$$

$$0 \leq g^a < 1, \tag{25}$$

$$0 \leq g^s < 1, \tag{26}$$

where $(\boldsymbol{\beta}, \boldsymbol{\gamma})$ are distributed according to $p(\boldsymbol{\beta}, \boldsymbol{\gamma} \mid y, \xi)$ and $\eta \in (0, 1)$ is the desired confidence level.

We approximate the chance constraint eq. (24) using the same importance-sampling estimator from eq. (7). Specifically, we set

$$f(\boldsymbol{\beta}, \boldsymbol{\gamma}) = \mathbf{1}\big\{\lambda_{\max}\big(M(t_0, \boldsymbol{\beta}, \boldsymbol{\gamma}, g^a, g^s)\big) \leq -\alpha\big\},$$

and plug this into eq. (7) to obtain

$$\mathbb{P}\big(\lambda_{\max}\big(M(t_0, \boldsymbol{\beta}, \boldsymbol{\gamma}, g^a, g^s)\big) \leq -\alpha \,\big|\, y, \xi\big) \approx \sum_{i=1}^{N} \tilde{w}_i \, \mathbf{1}\big\{\lambda_{\max}\big(M(t_0, \boldsymbol{\beta}_i, \boldsymbol{\gamma}_i, g^a, g^s)\big) \leq -\alpha\big\},$$

where the normalized weights $\tilde{w}_i$ are as in the main-text definition eq. (7). This provides a numerically stable way to evaluate the chance constraint and solve the resulting optimization problem using semidefinite programming.

### E.1.2 CVAR-BASED ROBUST CONTROL

As an alternative to the scenario-based chance constraint in Appendix E.1.1, we can control the *tail* of constraint violations using Conditional Value-at-Risk (CVaR).

Let the feasibility event be

$$\mathbb{P}\big(\lambda_{\max}\big(M(t_0, \boldsymbol{\beta}, \boldsymbol{\gamma}, g^a, g^s)\big) \leq -\alpha \,\big|\, y, \xi\big) \geq \eta.$$

For posterior (or variational) samples $\{(\boldsymbol{\beta}_i, \boldsymbol{\gamma}_i)\}_{i=1}^{N}$, define the per-sample violation

$$v_i(g^a, g^s) := \lambda_{\max}\big(M(t_0, \boldsymbol{\beta}_i, \boldsymbol{\gamma}_i, g^a, g^s)\big) + \alpha.$$

We enforce $\mathrm{CVaR}_{\eta}(v) \leq 0$ using the Rockafellar–Uryasev sample-average epigraph formulation with normalized weights $\bar{w}_i$ (default $\bar{w}_i = 1/N$):

$$s_i \geq v_i(g^a, g^s) - \tau, \quad i = 1, \ldots, N, \qquad \tau + \frac{1}{1-\eta} \sum_{i=1}^{N} \bar{w}_i \, s_i \leq 0, \tag{27}$$

with decision variables $\tau \in \mathbb{R}$ and $s_i \geq 0$. At optimality, $s_i = (v_i(g^a, g^s) - \tau)_+$, so only the upper tail beyond the $\eta$-quantile contributes.

Our robust controls are then obtained by solving the convex program

$$\begin{aligned}
\min_{g^a, g^s, \tau, \{s_i\}} \quad & J(g^a, g^s) \\
\text{s.t.} \quad & s_i \geq v_i(g^a, g^s) - \tau, \quad s_i \geq 0, \quad i = 1, \ldots, N, \\
& \tau + \frac{1}{1-\eta} \sum_{i=1}^{N} \bar{w}_i \, s_i \leq 0, \\
& 0 \leq g^a < 1, \quad 0 \leq g^s < 1,
\end{aligned} \tag{28}$$

where $J(g^a, g^s)$ is the same convex objective as in Equation (23). The specific form of $v_i$ is model dependent; in our SIQR case it is implemented via a convex epigraph for the spectral violation, so eq. (28) remains a tractable semidefinite program.

### E.2 PK MODEL

**Parameter influence.** For the PK model, the peak concentration $C_{\max}$ scales roughly as $D/V$ and increases with $k_a$ (faster absorption, smaller $t_{\max}$), decreases with $k_e$ (faster elimination), and is sensitive to the ratio $k_a/k_e$ (flip–flop kinetics when $k_a < k_e$).

We follow the same design–control split as in the SIQR example. The design $\xi$ (e.g., blood sampling schedule and/or formulation choice) is used to learn the PK parameters $\boldsymbol{\theta} = (k_a, k_e, V)$ via the posterior $p(\boldsymbol{\theta} \mid y, \xi)$. The control variable $g \in [0, 1]$ is a dose fraction, with the administered dose given by $D(g)$ (e.g., $D(g) = gD_0$ or a more general convex mapping). All exposure and risk quantities below depend on $g$ through $D(g)$ and on $\boldsymbol{\theta}$.

Posterior expectations in the PK setting are computed using the same importance-sampling estimator eq. (7), now with $\boldsymbol{\theta}$ in place of $(\boldsymbol{\beta}, \boldsymbol{\gamma})$.

#### E.2.1 CHANCE-CONSTRAINED CONTROL (PK ANALOGUE OF SIQR)

Choose a convex cost $J(g)$ (e.g., $J(g) = c\,g$ to discourage high dosing). The chance-constrained PK problem mirrors the SIQR formulation:

$$\min_{0 \leq g \leq 1} \quad J(g) \tag{29}$$
$$\text{s.t.} \quad \mathbb{P}(C_{\max}(g, \boldsymbol{\theta}) \leq C_{\text{thresh}} \mid y, \xi) \ \geq \ \eta,$$
$$\mathbb{E}[\text{AUC}(g, \boldsymbol{\theta})] \ \geq \ \text{AUC}_{\min}.$$

Both the probability and the expectation are evaluated using eq. (7), with $f(\boldsymbol{\theta}) = \mathbf{1}\{C_{\max}(g, \boldsymbol{\theta}) \leq C_{\text{thresh}}\}$ and $f(\boldsymbol{\theta}) = \text{AUC}(g, \boldsymbol{\theta})$, respectively.

#### E.2.2 CVAR-ROBUST CONTROL (EPIGRAPH FORM)

Define the per-sample toxicity violation (optionally with a nonnegative margin $\alpha$) as

$$v_i(g) = C_{\max}(g, \boldsymbol{\theta}_i) - C_{\text{thresh}} + \alpha, \qquad i = 1, \dots, N, \tag{30}$$

with normalized importance weights $\tilde{w}_i$ computed from eq. (7). We enforce $\text{CVaR}_\eta(g) \leq 0$ via

$$\min_{0 \leq g \leq 1, \, \tau, \, s_i \geq 0} \quad J(g) \tag{31}$$
$$\text{s.t.} \quad s_i \ \geq \ v_i(g) - \tau, \qquad i = 1, \dots, N,$$
$$\tau + \frac{1}{1 - \eta} \sum_{i=1}^{N} \tilde{w}_i \, s_i \ \leq \ 0,$$
$$\mathbb{E}[\text{AUC}(g, \boldsymbol{\theta})] \ \geq \ \text{AUC}_{\min}.$$

Here the expectation in the AUC constraint is again evaluated using eq. (7). This yields a PK control policy that trades off dose cost against a tail-robust toxicity constraint and a minimum-exposure requirement.

## F NUMERICAL DETAILS

### F.1 SIQR EXPERIMENT SETUP

For the SIQR model, we follow the parameterization framework of Talaei et al. (2024). We place independent log-normal priors on the transmission and recovery rates (in units of counts per day). In log-space, the transmission rates for asymptomatic and symptomatic individuals, $\beta^a$ and $\beta^s$, and the recovery rates $\gamma^a$ and $\gamma^s$ are log-normally distributed as

$$(\log \beta^a, \log \beta^s, \log \gamma^a, \log \gamma^s) \sim \mathcal{N}\big((0.5,\, 0.8,\, 0.2,\, 0.2),\, \text{diag}(0.5^2, 0.5^2, 0.3^2, 0.3^2)\big).$$

The stability margin is fixed at $\alpha = 0.05$, and the economic cost parameters are set to $(z^a, z^s) = (0.4,\, 0.6)$.

The design variable $\xi \in [1, 100]$ denotes the observation day. At time $\xi$ we observe $y_{\text{obs}} = (y_{\text{obs}}^a, y_{\text{obs}}^s)$, the counts of asymptomatic and symptomatic infected individuals. We model these

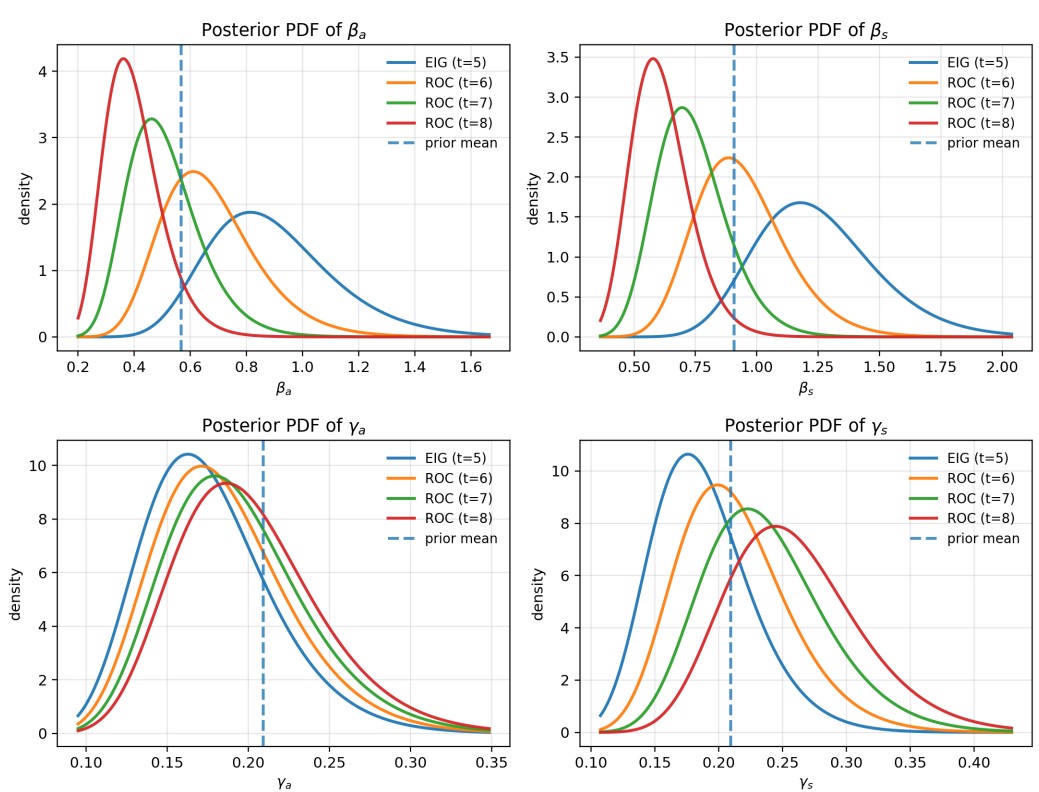

Figure 5: posterior densities for the SIQR model parameters $(\beta_a, \beta_s, \gamma_a, \gamma_s)$ under different observation times.

observations with a Poisson likelihood whose rate is a slightly perturbed version of the model prediction,

$$y_{\text{obs}} \mid \xi \sim \text{Poisson}(\lambda(\xi)), \qquad \lambda(\xi) = 0.95\, y_{\text{true}}(\xi),$$

where $y_{\text{true}}(\xi)$ is the SIQR solution evaluated at time $\xi$ under a given parameter draw. The amortized variational posterior $q_\phi(\boldsymbol{\beta}, \boldsymbol{\gamma} \mid y, \xi)$ is trained on simulated pairs $(y, \xi)$ as described in Appendix B.2, and the resulting posterior is used inside the robust control layer. The convex optimization problems for quarantine rates $(g^a, g^s)$ are solved via a semidefinite programming formulation (see Appendix D for details).

### F.2 PK EXPERIMENT SETUP

For the PK model, we follow the parameterization in Kleinegesse & Gutmann (2021). We place independent log-normal priors on the absorption rate $k_a$, elimination rate $k_e$, and volume of distribution $V$. In log-space, the parameters satisfy

$$(\log k_a, \log k_e, \log V) \sim \mathcal{N}\big((0,\ \log 0.1,\ \log 20.0),\ \text{diag}(0.05^2, 0.05^2, 0.05^2)\big).$$

The Bateman PK model, the definitions of $C_{\max}$ and AUC, and the associated robust control formulations (chance constraints and CVaR) are given in Appendices C.3 and E. The same amortized variational inference and importance-weighted posterior estimation pipeline is used as in the SIQR experiments, with the control variable $g$ parameterizing the administered dose $D(g)$.

### F.3 ADDITIONAL POSTERIOR VISUALIZATION RESULT

This appendix provides an additional posterior visualization for the SIQR model. Figure 5 shows the marginal posterior densities of the SIQR parameters. The blue curves correspond to the EIG-optimal

design at $t = 5$, while the orange, green, and red curves correspond to GoBOED designs optimized for robust optimal control at $t = 6$, $t = 7$, and $t = 8$, respectively. The dashed vertical line denotes the prior mean, which is also used as the data-generating value in the forward model, and hence as the reference point for all posterior estimates. Relative to the EIG design, the GoBOED-oriented designs yield posteriors that are more concentrated and systematically shifted for the transmission and recovery rates, indicating that GoBOED favors parameter configurations that most strongly influence the control policy.

## G    EXTENDED RELATED WORK

**Goal-oriented Bayesian optimal experimental design** Recent advances in BOED have shifted focus from parameter estimation to optimizing for specific quantities of interest (QoIs) — measurable outcomes that directly impact decision-making. For linear models, Attia et al. (2018) established the framework for goal-oriented optimal design of experiments (GOODE) that simplifies computational evaluation for experimental design. Building on this work, Neuberger et al. (2024) introduced a "$G_q$-optimality" criterion based on quadratic approximation of goal functionals for PDE-governed linear inverse problems. Additionally, for Bayesian linear inverse problems, Madhavan et al. (2025) developed a control-oriented approach that connects optimal control and sensor placement while prioritizing uncertainty reduction in controlled state variables. The linearity in these models makes the problems mathematically tractable and computationally efficient to solve. However, many real-world systems, including epidemic models, exhibit significant nonlinearities that require more sophisticated approaches.

For handling non-linear models, Zhong et al. (2024) created a computational framework using nested Monte Carlo estimators, Markov chain Monte Carlo (MCMC), kernel density estimation, and Bayesian optimization to address both non-linear observation models and prediction models. Similarly, Bickford Smith et al. (2023) proposed the expected predictive information gain (EPIG), an acquisition function that measures information gain in the space of predictions rather than parameters. Taking a different approach, Huang et al. (2024) introduced a decision-aware framework with a transformer neural decision process that simultaneously generates experimental designs and infers decisions in a unified workflow. For causal discovery problems, Tigas et al. (2022) developed methods to optimize intervention timing for large nonlinear structural causal models. Collectively, these works represent a paradigm shift toward experimental designs that optimize directly for decision-relevant outcomes rather than intermediate parameter estimates.

**Bayesian decision theory** Bayesian decision theory, which applies observed data to update posterior distributions for optimal decision-making, was formalized in Chaloner & Verdinelli (1995). Building on this foundation, Lacoste–Julien et al. (2011) developed a method that calibrates approximate inference techniques according to specific decision tasks using the Expectation-Maximization algorithm. For modern machine learning applications, Krishnan & Tickoo (2020) introduced a differentiable approach that balances accuracy against uncertainty calibration, enabling models to learn well-calibrated uncertainties while improving performance. Addressing computational efficiency challenges, Gordon et al. (2018) developed a framework that uses few-shot learning to simplify posterior inference of task-specific parameters, eliminating the need for gradient-based optimization during testing. These advances have progressively made Bayesian decision-making more practical for complex problems with computational constraints.

**Robust decision-making** With the growing interest in goal-oriented BOED, robust decision-making has been studied in many application domains. For example, compartmental network-based approaches (e.g., SIQR model) are widely adopted in epidemic management. Two main control strategies dominate current research: optimal control to minimize infection rates (Lee et al., 2010; Hayhoe et al., 2021; Khanafer & Başar, 2014; Liu & Buss, 2020; Bock & Jayathunga, 2018) and spectral optimization for resource allocation (Hota et al., 2021; Mai et al., 2018; Smith & Bullo, 2023; Preciado et al., 2014; Enyioha et al., 2015). A significant challenge with these approaches is their computational complexity, as many of the underlying problems are NP-complete or NP-hard (Mieghem et al., 2011). In a parallel vein, PK models play a crucial role in optimizing drug dosing and improving patient outcomes by quantitatively linking individual variability to clinical efficacy and safety (Agema et al., 2025; Lai et al., 2022). These models help guide dose selection and treatment personalization,

especially under uncertainty in drug absorption, metabolism, and patient response (Zavřelová et al., 2025; Norris, 2023).

