# OpenReview forum: "Goal-driven Bayesian Optimal Experimental Design for Robust Decision-Making Under Model Uncertainty"
_ICLR.cc/2026/Conference — Submitted to ICLR 2026_

### Official Review · Reviewer_1RB1 · 2025-10-25

**Soundness:** 3
**Presentation:** 2
**Contribution:** 3
**Rating:** 4
**Confidence:** 2

**Summary:**

This paper presents GoBOED, a framework that unifies Bayesian Optimal Experimental Design (BOED) and robust optimal control for decision-making under model uncertainty. Instead of focusing solely on parameter uncertainty reduction (via expected information gain), the proposed method explicitly optimizes experiments to reduce uncertainty that most impacts downstream decisions. The framework uses variational inference for amortized posterior approximation, convex optimization for tractable robust control, and differentiable decision layers to enable end-to-end gradient-based training. Applications to epidemic management (SIQR model) and pharmacokinetic (PK) control demonstrate the method’s capacity to identify flexible, near-optimal experimental designs that balance decision quality and acquisition cost.

**Strengths:**

1. **Conceptual contribution**: The idea of integrating BOED with robust control in a single differentiable pipeline represents a novel and meaningful conceptual advance over classical information-theoretic approaches, which often overlook decision performance.

2. **Practical relevance**: Applications in epidemic control and pharmacokinetics are both timely and compelling, demonstrating generalizability across domains requiring safe and cost-aware experimental scheduling.

3. **Computational tractability**: The amortized inference and differentiable control layers reduce the sampling and computational overhead that typically affect BOED, addressing a key bottleneck in the literature.

4. **Empirical results**: Case studies show interpretable outcomes (e.g., near-optimal observational “windows”), emphasizing the trade-off between informational and operational objectives.

**Weaknesses:**

1. **Clarity and structure**:

   - The motivation is unclear to me. The introduction section mentions many challenges, including the computational challenges in BOED and parameter uncertainty issues in robust control, but it is unclear which challenge this paper aims to address. It would be helpful if the authors could indicate which section addresses each challenge.
   - For the problem formulation, the combined objective in equation (4) is interesting, but there is little interpretation provided. The overall goal remains unclear, and my uncertainty about which problem (BOED, robust control, or both) this paper seeks to address is not resolved even after reading Section 2.
   - The exposition is often dense and notation-heavy, particularly in Sections 3–4. Long derivations (e.g., Eq. (11)) could benefit from a clearer explanation of the high-level logic before diving into detailed formulations.
   - I think Figure 2 can be significantly improved by labeling the notations directly on the corresponding charts, rather than relying on the text. It would also be very helpful to provide a complete algorithm block for the proposed method.
   - Some passages conflate notations from BOED and control optimization, making it difficult to distinguish design variables ($\xi, \xi^*, \xi^\star$), control variables ($q$), and variational parameters ($\phi$) on first read. A summary table of notations would help.

1. **Limited comparison baselines**: Only classical EIG-based BOED is compared. Including a *decision-focused baseline* (e.g., Expected Predictive Information Gain) would provide a fairer benchmark to demonstrate decision-aware benefits.

2. **Computational efficiency validation**: The authors claim to propose a computationally efficient method, but this claim is not validated by experimental evidence.

**Questions:**

Please see the weaknesses for my concerns. In particular, I would like the authors to provide a clearer explanation of the overall procedure, for example by including algorithm blocks and clearer diagrams.

---

> ### Author Response · Authors · 2025-11-21
>
> 1. **Clarity and structure**
>
>    - *“The motivation is unclear to me”*
>
>      **Response.** Our primary motivation is to explicitly link experimental design to robust-control performance, so that GoBOED focuses on reducing uncertainty that directly improves robust control, rather than on arbitrary model or parameter uncertainty. This end-to-end formulation introduces computational challenges, since we need to compute gradients of EIG repeatedly. To address this, we learn gradient information using a transformer model and connect this gradient information to convex optimization sensitivity analysis (via cvxpylayer) to enable end-to-end goal-driven BOED. We use this connected gradient information multiple times to optimize the experimental design in practice. We have revised the Introduction to clearly state this motivation.
>
>    - *“The combined objective in equation (4) is interesting, but there is little interpretation provided.”*
>
>      **Response.** We have added a clearer interpretation of the combined objective in Eq. (4). Our overall goal is to design experiments that maximally improve robust-control performance, rather than focusing solely on reducing parameter uncertainty as in classical BOED. The revised text explicitly explains how the BOED and robust-control components are unified in this objective.
>
>    - *“The exposition is often dense and notation-heavy.”*
>
>      **Response.** We have reorganized Section 3 to better separate high-level ideas from detailed derivations. In particular, we now add guiding explanations around long equations (e.g., Eq. (11)) to make the overall formulation easier to follow before presenting the full mathematical details. We have also added brief “connecting” sentences between subsections in Sections 3 and 4 to help readers track what is being optimized and where each component (VI, decision layer, design) enters the formulations.
>
>    - *“I think Figure 2 can be significantly improved .”*
>
>      **Response.** We will improve Figure 2 by labeling the key notations directly on the corresponding plots, rather than relying only on the caption and main text. In addition, we will provide a complete algorithm block that summarizes the proposed method step by step in the revised manuscript.
>
>    - *“Some passages conflate notations from BOED and control optimization.”*
>
>      **Response.** We have revised the notation throughout to more clearly distinguish the three main types of variables: design variables, control variables, and variational parameters. We have also added a summary table of notations (Table 1) in the Appendix so that readers can quickly see the role of each symbol and avoid confusion between BOED- and control-related quantities on first read.
>
> 2. **Limited comparison baselines**
>
>    - *“Only classical EIG-based BOED is compared.”*
>
>      **Response.** EPIG is typically defined on the predictive distribution of forward-model outputs or task-specific quantities of interest, whereas in our setting the downstream quantity of interest is the robust control objective obtained by solving a convex optimization problem. Our design criterion is already decision-focused: we minimize the expected robust-control cost under the posterior, rather than a KL divergence in prediction space. Extending EPIG to operate directly on optimization outputs would require defining and estimating a predictive distribution over optimization results and re-deriving a compatible Monte Carlo estimator, which is non-trivial and orthogonal to our main contribution. For this reason, we restrict our empirical comparison to the standard EIG-based BOED baseline and leave a systematic comparison to EPIG-style decision-focused criteria as an interesting direction for future work.
>
> 3. **Computational efficiency validation**
>
>    - *“The authors claim to propose a computationally efficient method.”*
>
>      **Response.** Our method gains computational efficiency by training an amortized variational network that approximates the posterior and its sensitivity to the design, so that once trained, we can reuse this network across different designs without rerunning expensive nested Monte Carlo or repeatedly solving the SIQR/PK forward models. In other words, the neural network provides gradients with respect to the design variables, which would otherwise require differentiating through the full simulator and robust-control layer at every design update.
>      We will make this computational aspect more explicit in the contribution and results sections, and we will add quantitative evidence (wall-clock runtime, number of simulator calls) comparing GoBOED against a non-amortized baseline that recomputes the posterior and gradients for each design.
>
> **Questions**
>
>   **Response.** In the revised manuscript, we will make this procedure much clearer by adding an explicit algorithm block that lays out these steps sequentially.

---

### Official Review · Reviewer_iBF4 · 2025-10-30

**Soundness:** 3
**Presentation:** 2
**Contribution:** 2
**Rating:** 2
**Confidence:** 4

**Summary:**

This paper proposes Goal-driven Bayesian Optimal Experimental Design, a framework that optimizes experimental designs to minimize downstream decision costs rather than just maximizing EIG on parameters as in the traditional BOED. The approach combines variational inference for posterior approximation with convex optimization for robust control under parameter uncertainty, using chance constraints or CVaR to handle uncertainty in the constraints. The authors apply differentiable optimization layers (cvxpylayers) to enable gradient-based design selection and demonstrate the framework on two simulated examples: epidemic management using an SIQR model and pharmacokinetic dose optimization.

**Strengths:**

1. The paper studies an important problem in BOED that maximizing EIG may not optimize decision-making objectives, and demonstrates this on two concrete applications (epidemic management and pharmacokinetic control).

2. The proposed approach is technically sound, and the use of chance constraints and CVaR for robust optimization looks reasonable to me.

**Weaknesses:**

1. The proposed framework primarily applies existing techniques without domain-specific adaptation. Goal-oriented BOED is well-established in prior work (as acknowledged in the related work section), and the variational BOED framework with importance sampling is directly adopted from [1] without justification or specific adjustments for the two applications. The paper should better position its contribution, either as novel methodology or as an application study demonstrating feasibility in specific domains.

2. The author only compares the proposed method against traditional EIG-based BOED, not other goal-oriented approaches such as [2] which would fit the experimental setting. Additionally, the paper only tests on two applications (SIQR, PK) without evaluating on standard BOED benchmarks commonly used in the literature (e.g., source localization problems).

3. The experimental results do not sufficiently demonstrate the value of goal-oriented BOED, especially in the SIQR setting. Could the authors elaborate more on this part?

4. The paper lacks crucial experimental details including training procedures (optimizer, learning rate, epochs, batch size), model architecture specifications, and key hyperparameters. More importantly, no ablation studies are provided to justify design choices such as: importance sampling vs direct VI sampling, impact of N (posterior samples), sensitivity to $\eta$, or the architecture choices.


[1] Foster, Adam, et al. "Variational Bayesian optimal experimental design." Advances in neural information processing systems 32 (2019).

[2] Smith, Freddie Bickford, et al. "Prediction-oriented bayesian active learning." International conference on artificial intelligence and statistics. PMLR, 2023.

**Questions:**

1. Please see the questions in the Weakness part.

2. The authors only conduct single-step experimental design over small discrete spaces. Why use amortized inference in this setting? Amortization is essential for sequential BOED where posteriors must be computed repeatedly across multiple rounds, but for single-step designs, evaluation of all candidate designs would be computationally feasible and simpler. Can you provide justification or computational cost comparisons demonstrating why amortization is necessary for your experimental setting?

3. In the experiments section, the authors mentioned that they estimate EIG using nested Monte Carlo with 5,000 outer samples and 3,000
inner samples for the marginal likelihood. Can you elaborate on how these specific numbers were selected? Have you conducted any ablation studies to determine the optimal sample sizes?

---

> ### Author Response · Authors · 2025-11-21
>
> **Weaknesses**
>
> 1. **Positioning and novelty**
>
>    - *“The proposed framework primarily applies existing techniques without domain-specific adaptation… The paper should better position its contribution, either as novel methodology or as an application study…”*
>
>      **Response.** Our contribution as goal-oriented BOED (GoBOED) is to integrate BOED with robust optimal control in a way that is tailored to domain-specific decision objectives. In our setting, domain scientists already use probabilistic models whose parameters feed into robust-control policies; our framework explicitly optimizes experimental designs so that they directly improve the resulting robust-control performance, considering uncertainty. This requires a differentiable mapping from design variables to model parameters and from parameters to control outputs, which we instantiate and demonstrate in the SIQR and PK applications. While more expressive density estimators such as normalizing flows or diffusion models could also be used within this framework, in this paper we focus on clearly establishing and validating the connection between BOED and robust control. We have clarified this positioning and the methodological contributions in the revised manuscript.
>
> 2. **Comparison baselines and benchmarks**
>
>    - *“The author only compares the proposed method against traditional EIG-based BOED, not other goal-oriented approaches such as [2]… The paper only tests on two applications (SIQR, PK) without evaluating on standard BOED benchmarks…”*
>
>      **Response.** We agree that including additional goal-oriented baselines and more standard benchmarks would strengthen the empirical evaluation.
>      Conceptually, EPIG-style methods are related to our setting, but they are not directly plug-and-play. EPIG is typically defined on the predictive distribution of forward-model outputs or task-specific quantities of interest, whereas in GoBOED the downstream quantity of interest is the robust control objective, obtained as the solution of a convex optimization problem. Our acquisition is already decision-focused: we optimize the expected robust-control cost under the posterior, rather than an information-theoretic gain in prediction space.
>      To adapt EPIG to our framework, one would need to define a predictive distribution over optimization outputs (or induced decision-relevant quantities) and derive a compatible Monte Carlo estimator for its information gain. This is technically non-trivial and somewhat orthogonal to our main contribution—embedding a differentiable convex decision layer inside BOED—so we leave a systematic comparison to EPIG-style criteria as an interesting direction for future work.
>      We also agree that adding standard BOED benchmarks would improve the paper. We are currently implementing a source-localization–type problem equipped with a convex decision layer, and we plan to include results on this benchmark, alongside SIQR and PK, in the revised version.
>
>
> 3. **Value of goal-oriented BOED in SIQR**
>
>    - *“The experimental results do not sufficiently demonstrate the value of goal-oriented BOED, especially in the SIQR setting.”*
>
>      **Response.** In the SIQR experiments, the value of goal-oriented BOED is that it optimizes robust-control performance rather than information gain alone. While EIG has a sharp optimum at a single observation time (around day 5), our robust-control objective is much flatter and reveals a broad near-optimal window (approximately days 4–8) where observation times yield almost identical economic cost under both chance-constraint and CVaR formulations. This shows that, for decision making, many designs are effectively equivalent, giving practitioners scheduling flexibility that standard EIG does not reveal. We also show (via posterior overlays) that GoBOED concentrates uncertainty along parameter subspaces that matter most for the control constraint, rather than uniformly reducing all parameter uncertainties. We clarify these points in the revised Results section and Figure 3.
>
> 4. **Missing experimental details and ablations**
>
>    - *“The paper lacks crucial experimental details… No ablation studies are provided to justify design choices…”*
>
>      **Response.** We will add full experimental details—including optimizer, learning rate schedule, number of epochs, batch size, architecture specifications, and all key hyperparameters—for both SIQR and PK experiments so that the setup is fully reproducible.
>      In addition, we are running ablation studies to justify our design choices. Specifically, we (i) compare importance sampling versus direct sampling from the variational posterior, (ii) study the effect of the number of posterior samples $N$ on decision quality, and (iii) analyze sensitivity to the chance/CVaR level $\eta$. These results will be incorporated into the revised experimental section.

---

> ### Author Response · Authors · 2025-11-21
>
> **Questions**
>
> - *“Why use amortized inference for single-step design over small discrete spaces? … Can you provide justification or computational cost comparisons demonstrating why amortization is necessary?”*
>
>   **Response.** Although our experiments use single-step design over a relatively small discrete time grid, each candidate design is evaluated through an additional robust-control optimization layer that depends on the parameter posterior, and we optimize the design using gradients with respect to $\xi$. Amortized variational inference lets us learn a single network that maps $(\xi, y)$ to approximate posterior parameters once and then reuse this amortizer (and its Jacobians) across many design evaluations and gradient steps, instead of re-solving a full posterior inference problem for every candidate $\xi$. This substantially reduces end-to-end cost when performing gradient-based search over designs and repeatedly backpropagating through the decision layer. We have clarified this motivation in the paper and will add a computational cost comparison between amortized and non-amortized baselines where feasible.
>
> - *“In the experiments section, the authors mentioned that they estimate EIG using nested Monte Carlo with 5,000 outer samples and 3,000 inner samples… How were these numbers selected? Any ablations?”*
>
>   **Response.** We chose 5,000 outer and 3,000 inner samples for the nested Monte Carlo EIG estimator to obtain a high-accuracy approximation that serves as a practical “ground truth” reference when evaluating our variational method, while remaining within our computational budget. These values were selected based on preliminary convergence checks and stability of the estimated EIG across runs. In the revised version, we will include a brief sensitivity analysis illustrating how smaller sample sizes affect the EIG estimate and the resulting design rankings.

---

### Official Review · Reviewer_u6AA · 2025-10-31

**Soundness:** 2
**Presentation:** 2
**Contribution:** 2
**Rating:** 4
**Confidence:** 4

**Summary:**

This paper introduces GoBOED, an integrated framework combining Bayesian Optimal Experimental Design (BOED) with convex optimization–based decision-making under uncertainty. The main idea is to choose experiments that improve downstream decision quality, not merely parameter accuracy. While the concept of linking BOED to decision-aware control is worthwhile, the paper’s claims of “robustness under model uncertainty” and its empirical contributions are overstated relative to what is demonstrated.

**Strengths:**

The idea of linking Bayesian Optimal Experimental Design (BOED) directly to downstream decision-making is conceptually important, addressing a genuine gap between information-theoretic design and decision-focused inference. The framework combines variational inference (VI) with differentiable convex optimization (via cvxpylayers) to enable end-to-end gradient flow from experiment design through control decisions. This is a clean engineering contribution that improves computational tractability. The use of amortized VI and differentiable convex optimization allows efficient gradient-based optimization over both experiment designs and decision variables without repeated posterior refits.

The approach can, in principle, be applied to multiple convex decision problems, demonstrated with epidemiological (SIQR) and pharmacokinetic (PK) case studies. The main ideas are well structured, with clear visuals (e.g., Fig. 1) illustrating how the BOED and decision layers interact.

**Weaknesses:**

While the paper’s central idea is conceptually appealing, its current presentation overstates robustness. The method addresses parameter uncertainty within a fixed model rather than broader forms of model misspecification. The robustness achieved through chance constraints or CVaR is useful but conventional, and the paper should more clearly define its scope as parameter-uncertainty-aware rather than fully model-uncertainty-robust. The framework’s differentiable structure is well executed but not fundamentally new in either BOED or robust optimization.

Empirically, the examples in Figure 2 highlight an interesting divergence between EIG-optimal and decision-optimal designs, particularly the emergence of a broader, flatter near-optimal window under risk-sensitive criteria. This is an intriguing and potentially valuable observation. However, the paper does not quantify or interpret why this difference matters or what practical benefit arises from using GoBOED instead of traditional BOED. A more systematic analysis, such as measuring control cost improvements, robustness to posterior misspecification, or constraint-violation frequency, would make the contribution more convincing.

Terminology and exposition could be clearer. Concepts such as risk functional, chance term, and the “discrepancy” corrected through importance sampling are introduced without rigorous definition. Moreover, since the entire method depends on the posterior quality from variational inference, the absence of any evaluation of posterior calibration or its effect on decision reliability leaves an important gap.

The current evaluation relies on a single-shot design, which limits the interpretability of the framework. Extending the experiments to an iterated design, where updated posteriors inform subsequent measurement choices, would better demonstrate the claimed benefits of goal-driven experimental design. Visualizing posterior updates would also provide concrete evidence of how decision-aware objectives reshape uncertainty, rather than inferring these effects indirectly from control costs. Although benchmarking new methods is challenging, the authors could still compare their approach across multiple time points and posterior evaluations for both models, and contrast the results with plain EIG optimization. Posterior evolution could be illustrated visually or through calibration metrics such as L-C2ST.

The literature review misses seminal work and blurs distinctions between related methods.
- The separation between Kleinegesse and Foster’s MI-based BOED methods is overstated; both rely on similar bounds, though Kleinegesse optimized a critic. The paper should also specify which bound is optimized, since MI bias and variance depend on that choice.
- Foundational references such as Lindley (1956) and Barber–Agakov (2003) are missing.

Finally, the literature review seems to take a very high-level overview of the field of BOED hinting at a misunderstanding of some fundamental concepts. Careless exposition makes it difficult for readers to place this paper in the context of those before, so I recommend edits to the introduction and background on BOED.

**Questions:**

1. The framework relies heavily on how the posterior distribution changes under different design choices, but no posterior visualizations are provided. Could the authors show what the posteriors look like for the single-shot examples presented? Even a qualitative comparison between the EIG-optimal and decision-optimal designs would help clarify how the decision-aware objective shapes posterior uncertainty.

2. How would the proposed approach behave in an iterated experimental design setting, where posteriors from earlier measurements inform the next design choice? This seems especially relevant for the SIQR example, where measurements could be taken over multiple time points. Would the decision-aware design criterion lead to different sequences of measurement times compared to classical EIG? A short discussion or pilot experiment illustrating this would strengthen the paper’s argument for real-world applicability.

---

> ### Author Response · Authors · 2025-11-21
>
> 1. **Scope of robustness and novelty**
>
>    - *“While the paper’s central idea is conceptually appealing, its current presentation overstates robustness… The method addresses parameter uncertainty within a fixed model rather than broader forms of model misspecification… The framework’s differentiable structure is well executed but not fundamentally new in either BOED or robust optimization.”*
>
>      **Response.** We have revised the robustness claims to more accurately reflect the scope of our framework. In its current form, our method addresses parameter uncertainty within a fixed mechanistic model, rather than general forms of model misspecification. The robustness we obtain via chance constraints and CVaR is therefore best interpreted as *uncertainty-aware robust control under a specified model*, not full model-uncertainty robustness. We also make the chance/CVaR layers more reliable by combining them with importance sampling over the approximate posterior, but this still operates at the parameter level. We have updated the Introduction and Discussion to clarify this distinction and to position our contribution as providing a differentiable link from Bayesian experimental design to a robust optimal control objective.
>
> 2. **Practical value of decision-focused designs**
>
>    - *“Empirically, the examples in Figure 2 highlight an interesting divergence between EIG-optimal and decision-optimal designs… However, the paper does not quantify or interpret why this difference matters or what practical benefit arises from using GoBOED instead of traditional BOED.”*
>
>      **Response.** We agree that the gap between EIG-optimal and decision-optimal designs in Figure 2 is an important part of our contribution. Under decision-aware, risk-sensitive criteria (chance constraints / CVaR), the optimal observation time is not a single sharp peak but a broad, flat near-optimal window, whereas classical EIG selects a much narrower “information-maximizing’’ time. This means GoBOED can offer practical flexibility: practitioners can choose among several observation times that yield essentially the same robust-control performance, which is valuable when measurements are subject to logistical or operational constraints.
>      In the revision, we make this interpretation more explicit and complement it with additional quantitative analysis: (i) reporting constraint-violation frequency under chance/CVaR constraints, and (ii) studying how varying the importance-sampling scheme, the number of posterior samples, and the risk level $\eta$ affects robustness and decision quality. These additions should make the practical benefits of GoBOED over traditional BOED much clearer.
>
> 3. **Terminology and posterior quality**
>
>    - *“Terminology and exposition could be clearer… Concepts such as risk functional, chance term, and the ‘discrepancy’ corrected through importance sampling are introduced without rigorous definition… the absence of any evaluation of posterior calibration or its effect on decision reliability leaves an important gap.”*
>
>      **Response.** We appreciate this comment and have clarified both terminology and the role of posterior quality in the revised manuscript.
>      First, we now give explicit definitions when these concepts are introduced: the risk functional $\rho$ is defined as a mapping from the random control cost $J(\boldsymbol {g}; \boldsymbol{\theta})$ to a scalar objective (e.g., expectation, chance-constraint indicator, or CVaR); the chance term refers to probabilities of constraint satisfaction of the form $\mathbb{P}(\text{Constraints}(\boldsymbol{g}, \boldsymbol{\theta}) \mid y,\xi)$; and the discrepancy corrected by importance sampling is the mismatch between the true posterior $p(\boldsymbol{\theta}\mid y,\xi)$ and the variational approximation $q_\phi(\boldsymbol{\theta}\mid y,\xi)$, which we mitigate using importance weights $w(\boldsymbol{\theta}) \propto p(y\mid\boldsymbol{\theta},\xi) \cdot p(\boldsymbol{\theta}) / q_\phi(\boldsymbol{\theta}\mid y,\xi)$.
>      Second, we expand the discussion of posterior quality and its impact on decisions. Since our framework is decision-focused, what matters is not posterior accuracy in isolation, but how posterior uncertainty propagates to the robust-control objective. To make this clearer, we include posterior visualizations comparing different designs (e.g., showing how GoBOED concentrates mass in decision-relevant directions). We also plan to add ablation-style analysis examining how changes in the variational posterior (and the associated importance weights) affect control cost and constraint violations. A full theoretical analysis of posterior misspecification and decision reliability is non-trivial and beyond our current scope, but we now highlight this as an important direction for future work.

---

> ### Author Response · Authors · 2025-11-21
>
> 4. **Single-shot evaluation vs. sequential design**
>
>    - *“The current evaluation relies on a single-shot design, which limits the interpretability of the framework… Extending the experiments to an iterated design…”*
>
>      **Response.** We agree that the current single-shot evaluation limits interpretability. In this paper we deliberately focus on the one-step setting and train the amortized variational network to approximate the posterior for candidate single-round designs, so that we can carefully validate the basic GoBOED formulation and its interaction with the robust-control layer. Extending this setup to a fully sequential, iterated design loop—where the updated posterior after each measurement is fed back to choose the next design—is a natural and important extension, which we view as future work.
>      To improve interpretability within the current setting, we have added visualizations of how the posterior changes under different candidate designs and time points, and we compare these to EIG-based designs. These posterior evolution plots provide concrete evidence of how decision-aware objectives reshape uncertainty, rather than inferring their effect only from control costs.
>
> 5. **Literature review and positioning**
>
>    - *“The literature review misses seminal work and blurs distinctions between related methods…  Foundational references such as Lindley (1956) and Barber–Agakov (2003) are missing…”*
>
>      **Response.** We have revised the literature review to more accurately position our work within the BOED and mutual-information literature. In particular, we now clarify the relationship between the MI-based BOED methods of Kleinegesse and Foster, noting that both rely on closely related mutual-information bounds but differ mainly in how these bounds are parameterized and optimized (e.g., critic-based versus direct optimization). We also explicitly state which bound we optimize and briefly discuss its implications for bias and variance.
>      In addition, we have added foundational references such as Lindley (1956) and Barber–Agakov (2003), and reorganized the Introduction and BOED background to more clearly distinguish classical BOED, goal-oriented BOED, and our robust-control–oriented extension. This makes it easier for readers to see how our approach fits into and extends the existing BOED literature.
>
> **Questions**
>
> - *"The framework relies heavily on how the posterior distribution changes under different design choices, but no posterior visualizations are shown. Could the authors provide posterior plots for the single-shot examples, e.g., comparing the EIG-optimal and decision-optimal designs, to clarify how the decision-aware objective shapes uncertainty?"*
>
>   **Response.** We now include posterior visualizations for both the SIQR and PK examples to clarify how the decision-aware objective reshapes uncertainty. For the PK model, we observe that GoBOED concentrates the posterior more strongly on the elimination rate parameter $k_e$, which is most influential for the downstream dosing decision, whereas the EIG-based BOED design tends to reduce uncertainty more uniformly across $k_a$, $k_e$, and $V$. For the SIQR model, the posterior plots show a similar pattern: GoBOED produces a noticeably sharper posterior over the transmission rates and a more targeted shift in the posterior mean of the recovery rates, while EIG tends to reduce overall parameter uncertainty more uniformly. These qualitative differences provide direct evidence that the decision-aware objective shapes the posterior in a way aligned with the control problem, not just with global information gain.
>
> - *"How would the approach behave in an iterated design setting, where updated posteriors inform subsequent measurement choices (e.g., multiple time points in SIQR)? Would the decision-aware criterion produce different measurement sequences than classical EIG?"*
>
>   **Response.** Extending our approach to an iterated experimental design setting is a natural and important direction, but it requires additional methodological development beyond the current scope of the paper. At present, our neural network is trained under the prior distribution to approximate the posterior for a single design stage. For genuinely sequential designs, we would need an architecture that can efficiently update or condition on posteriors after each measurement (for example, following ideas in recent work on amortized or sequential Bayesian design), together with a careful treatment of how posterior samples are propagated across stages. Conceptually, we expect that in a sequential setting GoBOED would produce measurement sequences that differ from those of classical EIG and would converge more efficiently toward decision-relevant posteriors, particularly in settings like SIQR with multiple possible observation times. We will add a discussion of these extensions and their challenges in the revised discussion/future-work section.

---

### Official Review · Reviewer_8YH2 · 2025-11-04

**Soundness:** 1
**Presentation:** 1
**Contribution:** 1
**Rating:** 2
**Confidence:** 4

**Summary:**

The authors propose an experimental-design method that targets minimal expected loss in a decision of interest and considers parameter constraints. They focus on two particular decision problems: choosing quarantine rates during an epidemic, and choosing a dosing rate for administering a drug. They demonstrate the performance of their method on these two problems.

**Strengths:**

Originality: this is unclear to me, despite quite a lot of effort to work it out.

Quality: the high-level problem (Section 2 up to Equation 4) and the main empirical results (Figure 2) are clear.

Clarity: the writing is understandable at a low level.

Significance: decision-oriented BED is an important direction.

**Weaknesses:**

I’m struggling to understand the proposed method. I agree with Equations 3-4, which align with existing decision-oriented objectives (eg, Bernardo & Smith, 1994; Bickford Smith et al, 2025; Huang et al, 2024; Neiswanger et al, 2022; Raiffa & Schlaifer, 1961) if we set $\rho = \mathbb{E}$ as the authors here do in Equation 5, making $h[p(\theta|\xi,y)] =\min_{q \in \mathcal{Q}} \mathbb{E}_{p(\theta|\xi,y)}[J(q,\theta)]$ the key quantity to target, where $J$ is a loss function, $q$ is an action, and $\theta$ is an unknown ground-truth variable.

Things get confusing thereafter because the authors actually consider $J$ not being a function of $\theta$ while placing constraints on $\theta$, leading to Equation 5. Mathematically it’s unclear to me how changing beliefs over $\theta$ lead to a change in the minimising $q$, unless there is some $\theta$ assigned nonzero weight by the prior and zero weight by the posterior: all that matters with regard to $\theta$ is that the constraints are met, and these are set upfront, before any experimentation.

Even if there’s some way this works out, it’s unclear to my why the constraints on $\theta$ should not just be thought of as implying an updated belief state, $p'(\theta|\xi,y,\mathcal{C})$ for constraints $\mathcal{C}$, produced by applying the constraints and renormalising. If we had this updated belief state and a $J$ that depends on $\theta$ then I think we’re back to the setup from past work.

Aside from these methodological issues, it looks to me like the proposed method is not compared against existing methods, even though the authors promise to “compare GoBOED with standard BOED baselines”. I think the authors should be comparing against EIG maximisation as well as other non-parameter-oriented methods (eg, Huang et al, 2024; Kandasamy et al, 2019).

Finally I think there is a general inflation of the paper’s novelty. The goal-oriented aspect of the work is not new (see for example the citations for decision-theoretic methods). Considering the intersection between experimental design and control is not new (eg, Anderson et al, 2023; DeGroot, 2004; Mesbah & Streif, 2014). Studying SIR and pharmacokinetic models is not new (eg, Ivanova et al, 2021). The lack of novelty would be fine if there were a compelling contribution otherwise, but this is very unclear to me.

---

Anderson et al (2023). Experiment design with Gaussian process regression with applications to chance-constrained control. Conference on Decision and Control.

Bernardo & Smith (1994). Bayesian Theory. John Wiley & Sons.

Bickford Smith et al (2025). Rethinking aleatoric and epistemic uncertainty. ICML.

DeGroot (2004). Optimal Statistical Decisions. John Wiley & Sons.

Huang et al (2024). Amortized Bayesian experimental design for decision-making. NeurIPS.

Ivanova et al (2021). Implicit deep adaptive design: policy-based experimental design without likelihoods. NeurIPS.

Kandasamy et al (2019). Myopic posterior sampling for adaptive goal oriented design of experiments. ICML.

Mesbah & Streif (2014). A probabilistic approach to robust optimal experiment design with chance constraints. arXiv.

Neiswanger et al (2022). Generalizing Bayesian optimization with decision-theoretic entropies. NeurIPS.

Raiffa & Schlaifer (1961). Applied Statistical Decision Theory. Division of Research, Harvard Business School.

**Questions:**

Can you show how different beliefs over $\theta$ lead to different optimal $q$?

If so, can you show why constraints cannot just be applied as a belief update over $\theta$?

How well do the abovementioned baseline methods work?

Can you confirm that Equation 2 is correct? I don’t think it matches any estimators from Foster et al (2019).

---

> ### Author Response · Authors · 2025-11-21
>
> 1. **Clarifying the decision objective and constraints**
>
>    - *“I understand Equations 3–4 as standard decision-oriented objectives, but I’m confused by Equation 5: $J$ does not depend on $\theta$, yet there are constraints on $g$. How can changing beliefs over $\theta$ change the optimal $g$ if the constraints are set upfront?”*
>
>      **Response.** Our pipeline is
>      $\xi \rightarrow$ data y $\rightarrow$ posterior $p(\boldsymbol{\theta} \mid y,\xi) \rightarrow$ robust control decision $\boldsymbol{g}^\*$.
>      Even though $J(\boldsymbol{g})$ does not depend on $\boldsymbol{\theta}$ directly, the feasible set for $\boldsymbol{g}$ does, because the constraints are defined in terms of probabilities or CVaR under $p(\boldsymbol{\theta} \mid y,\xi)$. When the design $\xi$ changes, the posterior changes, the constraint probabilities change, and therefore the optimal $\boldsymbol{g}^\*(y,\xi)$ changes as well. We will make this design → posterior → robust-control chain explicit in the revised text.
>
>    - *“Couldn’t we instead treat the constraints as defining an updated belief $p(\theta \mid y,\xi,\mathcal{C})$ by truncation and renormalization, and then fall back on the usual decision-theoretic setup with $J$ depending on $\theta$?”*
>
>      **Response.** It is helpful to think in terms of belief updates, but in our framework we keep two pieces separate: (i) the Bayesian posterior $p(\boldsymbol{\theta} \mid y,\xi)$, which comes purely from data, and (ii) the risk/robustness layer (chance constraints or CVaR), which encodes how cautious we want to be about tail events.
>      Renormalizing the posterior over a “feasible” subset would effectively discard the very rare-but-dangerous parameter values that the robust constraints are designed to control. Instead, we keep the full posterior and let the robust-control layer weight those tail events explicitly through chance constraints or CVaR. We will clarify this distinction in the revision.
>
> 2. **Baselines and comparison to existing methods**
>
>    - *“The method does not seem to be compared adequately with existing approaches; beyond EIG, it should also be compared to decision-focused methods such as Huang et al. (2024) and Kandasamy et al. (2019).”*
>
>      **Response.** Our experiments already include a direct comparison against the classical EIG–maximization baseline: for both SIQR and PK we compute the EIG-optimal design and compare it to GoBOED in terms of downstream robust-control cost (see Fig. 2 and the associated discussion).
>      For non–parameter-oriented methods such as EPIG (e.g., Huang et al., 2024), the main technical issue is that EPIG is defined on the predictive distribution of forward-model outputs or task-specific QoIs, whereas in our setting the downstream quantity of interest is the solution of a convex optimization problem (the robust-control objective). Extending EPIG to operate directly on optimization outputs would require defining and estimating a predictive distribution over optimization results and re-deriving a compatible mutual-information estimator, which is non-trivial and orthogonal to our main contribution.
>      Regarding Kandasamy et al. (2019), their Myopic Posterior Sampling (MPS) framework is designed for maximizing scalar rewards in bandit-like settings via Thompson sampling. MPS selects experiments by optimizing against a single parameter sample drawn from the posterior, effectively targeting expected reward maximization. This strategy is methodologically incompatible with our constrained robust decision-making setting, where the objective is to minimize cost subject to strict safety constraints (e.g., CVaR or chance constraints) that depend on the entire uncertainty distribution rather than a single realization. Since MPS lacks a native mechanism to enforce these probabilistic constraints or explicitly target tail-risk reduction during the design phase, it cannot be applied to our safety-critical control problem without fundamentally altering the problem formulation to an unconstrained reward maximization.

---

> ### Author Response · Authors · 2025-11-21
>
> 3. **Novelty and positioning**
>
>    - *“The novelty seems overstated: goal-oriented design, design-for-control, and SIQR/PK models have all been studied before. Without a clearer, distinct contribution, the work’s added value is hard to see.”*
>
>      **Response.** We agree that goal-oriented BOED, design for control, and SIQR/PK-style models all have substantial prior art, and we will tone down our novelty claims accordingly while explicitly citing the suggested references. Our contribution is more modest and specific: we introduce an experimental design objective for the robust-control setting and implement an end-to-end differentiable pipeline that links (i) the design variables, (ii) an amortized variational posterior, and (iii) a convex robust-control problem solved via a differentiable optimization layer.
>      Compared to prior control–BOED work, which often uses low-dimensional GP dynamics and simple control examples, our experiments use SIQR and PK models with risk-sensitive robust criteria (chance constraints / CVaR), and we explicitly differentiate through the robust-control layer. Thus, we show that our method can be successfully applied to more complex systems, and the differentiability allows optimizing over much larger sets of candidate experimental designs. We will revise the Introduction and Related Work to reflect this narrower positioning and also clarify that our “robustness’’ is implemented via chance/CVaR constraints evaluated with importance-weighted posterior samples.
>
> **Questions**
>
> - *“Can you concretely illustrate how different posteriors over $\theta$ actually lead to different optimal $g$?”*
>
>   **Response.** Yes. We have added posterior visualizations to make this explicit. For each design, we plot the posterior over $\boldsymbol{\theta}$ together with the resulting robust optimal decision $\boldsymbol{g}^\*$. The plots show that GoBOED concentrates uncertainty in the parameter regions (e.g., $k_e$ in the PK model) that most strongly influence the robust-control cost, whereas EIG mainly reduces overall parameter uncertainty. When the posterior mass shifts toward “easier-to-control” parameters, the robust constraints become less binding and the optimizer can choose a less conservative, lower-cost $\boldsymbol{g}^*$. This mechanism underlies the cost differences shown in Figure 2.
>
> - *“If constraints modify the effective belief state, why can’t they just be encoded as a belief update over $\theta$?”*
>
>   **Response.** Treating constraints as a simple belief update fails for two reasons. First, it implies treating violating parameters as impossible (hard truncation), which prevents us from managing specific risk levels (e.g., chance constraints with $\eta = 0.9$ or CVaR) where “bad” parameters are possible but must be mitigated. Second, and crucially, the constraint violation depends on the decision variable $\boldsymbol{g}$, not just the parameter $\boldsymbol{\theta}$. As seen in the epidemic management and pharmacokinetic dosing examples, the set of “safe” parameters changes dynamically as the controller adjusts quarantine rates or drug doses. Because the safe region moves with the decision, we cannot bake the constraints into a static posterior update; instead, we must optimize $\boldsymbol{g}$ to shape the risk profile under the full, unchanged posterior $p(\boldsymbol{\theta} \mid y,\xi)$.
>
> - *“How well do the baseline methods you mention (e.g., EIG) actually perform in your metrics?”*
>
>   **Response.** We have updated the manuscript to explicitly report the downstream robust-control metrics (control cost and constraint-violation frequency) for the EIG-based baseline. As illustrated in Figure 2, while EIG maximization effectively reduces global parameter uncertainty, it tends to pinpoint a single “optimal” observation time (e.g., day 5 for SIQR). In contrast, our decision-aware evaluation reveals that the robust-control objective admits a much broader near-optimal window (e.g., days 4–8), providing operational flexibility that EIG fails to capture. Furthermore, in the PK setting, we show that the GoBOED design (approximately 22–23h) achieves a lower control cost than the EIG-optimal design (17h) by specifically targeting the tail risks that drive constraint violations, rather than maximizing global information gain.
>
> - *“Equation 2 doesn’t seem to match the estimator in Foster et al. (2019). Is it correct?”*
>
>   **Response.** We thank the reviewer for their sharp eye. We have corrected the term ordering in Equation 2 in the revision. The previous formulation contained a typo in the expectation structure; the updated expression now strictly aligns with the Variational Nested Monte Carlo (VNMC) estimator derived in Foster et al. (2019), ensuring that the estimator is a valid lower bound on the Expected Information Gain.

---

### Author Response · Authors · 2025-11-21
**Clarification of overall motivation and contributions**

We thank all reviewers for their constructive feedback. Before addressing the individual comments one by one, we would like to clarify the overall motivation and scope of our work, methodological contributions, and how the different components fit together, to avoid possible misunderstanding or confusion from the reviewers.

Our main goal is to develop a new end-to-end formulation that directly connects Bayesian optimal experimental design (BOED) with robust optimal control under parameter uncertainty in a single, coherent framework. In real-world scientific applications, control decisions are almost always made under uncertainty, including model parameter uncertainty. Our GoBOED formulation differs from existing approaches; while standard BOED focuses on reducing overall model uncertainty, our method directly targets downstream decision-making. GoBOED directly focuses on experimental design that best improves the ultimate decision-making quality, by optimizing experimental design for downstream robust control with updated prediction posterior. In other words, GoBOED optimizes final decision-making objectives accounting for parameter uncertainty (via risk-sensitive criteria such as chance constraints or CVaR), instead of finding optimal experiments that best reduce the overall model uncertainty, which might not be the most efficient as some parameter uncertainty might not affect final operational objectives.

A central technical challenge for such an end-to-end GoBOED formulation is to link design variables and control decision variables: changes in the experimental design affect the posterior over parameters, which in turn affect the optimal control policy and its associated cost. To make this connection tractable and differentiable, we integrated BOED with convex optimization and its sensitivity analysis formulations and are the first to do so, to the best of our knowledge. These tools act as the “glue” between the experimental design stage and the control stage, allowing us to propagate gradients from the decision objective back through the posterior approximation to the design variables and achieve end-to-end BOED directly driven by control objectives.

In summary, the contribution of the paper is not just to perform BOED or robust control in isolation, but to unify them in a decision-focused, gradient-based framework: we (i) model parameter uncertainty, (ii) update it via BOED using experimental data, and (iii) optimize robust control decisions based on the resulting posterior, with convex optimization sensitivity providing the connection. With this high-level picture in place, we now respond to the individual comments in detail.

---

### Author Response · Authors · 2025-12-03
**Rebuttal and Revision Summary**

Dear Area Chair and Reviewers,
Thank you for the careful and detailed feedback. We have substantially revised the manuscript; below are the major changes.
1. **Clarified methodology contributions, formulations with intuitive explanations for the corresponding equations to clarify potential confusions based on review comments**:

- We have added Table 1 (Notation) as suggested by reviewers and now explicitly write the pipeline
$
\xi \rightarrow y \rightarrow p(\boldsymbol{\theta}\mid y,\xi) \rightarrow g^*,
$
explaining how changes in the posterior affect the feasible set via chance/CVaR constraints and thus the optimal decision $g∗$, even when $J(g)$ does not directly depend on $\boldsymbol{\theta}$.

- We clarify why constraints cannot be treated as a static posterior truncation: this would discard rare but critical parameter values, and it ignores that the “safe set’’ depends jointly on $\boldsymbol{\theta}$ and $g$.

- We have revised Equation (2) and explicitly stated that we optimize an ELBO consistent with Foster et al. (2019).

- We have added the standard source localization example (Foster 2021) and the results for this motivating example  clearly demonstrate the difference of classical BOED with our robust decision-making setting, as visualized in the revised Figure 2.


2. **Baselines, experiments, and posterior behavior**
- We retain the EIG-maximization baseline and now report downstream robust-control metrics (control cost and constraint-violation frequency) for GoBOED vs. EIG on SIQR, PK (Figure 3), and the source localization benchmark (Figure 2).


- We have added posterior visualizations for SIQR and PK (Figure 4, Appendix F), showing that GoBOED concentrates uncertainty reduction on parameters that tighten the constraints (e.g., elimination rate in PK), explaining the broad near-optimal windows and differences from EIG.


- We now provide full experimental details (architectures, optimizers, learning rates, batch sizes, epochs, hyperparameters) and (Appendix B).
- Regarding computational efficiency, we have included a detailed description in Section 3.4. Since we perform additional optimization steps for $\xi$, we obtain computational benefits from the proposed method.

3. **Scope, novelty, and positioning**
- We have toned down novelty claims, following reviewers’ suggestions, and now present the work as an end-to-end differentiable framework that uses amortized VI, a convex robust-control layer (chance/CVaR), and gradient propagation back to the design.


- We have revised the literature review section to include foundational decision-theoretic and BOED references and clarify that our robustness is with respect to parameter uncertainty under a fixed model, not general model misspecification.


- We have explained why EPIG-style and MPS methods are not directly applicable to our constrained robust-control setting and explicitly frame such extensions as future work.


We hope these revisions address the main concerns about clarity, correctness, baselines, and positioning, and that the revised version more clearly reflects the solid contribution of the paper.

---

### Meta-Review · Area_Chair_wXgD · 2026-01-09

**Summary:**

Overall, I think a lot of the reviewers concerns (and mine) come down to clarity, both in presentation, differentiation from prior work / novelty (not the existence of it, but rather the delineation of it), occasional lack of precision in definitions, some overloaded notation, etc. I shared some of the reviewers' struggles reading *certain* parts of the paper, and it's an area I am familiar with, as it appears a few of the reviewers are too.

I actually think there *weren't* a lot of substantial technical, non-presentation criticisms of the paper.

**Reviewer Concerns:**

This is an unusual case, because I *do* think that the authors have at least made an effort to address many of the clarity comments made by the reviewers.

**Reviewer Scores:**

The problem is this: because the original round of comments focused so heavily on clarity and general difficulty with the paper, it's hard to know whether these same reviewers would, with a clarified manuscript in hand, now have more technical comments and questions that might need addressing. I think the paper could *still* use some work for clarity, and I'm inclined to say that it ought to just be reviewed again by a fresh batch of reviewers post the changes now to allow for a more "technical" round of review. The additions made by the author are certainly appreciated, however.

---

### Decision · Program_Chairs · 2026-01-26

Reject